



# Terpenes and their oxidation products in the French Landes forest: insight from Vocus PTR-TOF measurements

Haiyan Li[1], Matthieu Riva[2], Pekka Rantala[1], Liine Heikkinen[1], Kaspar Daellenbach[1], Jordan E. Krechmer[3], Pierre-Marie Flaud[4,5], Douglas Worsnop[3], Markku Kulmala[1], Eric Villenave[4,5], Emilie Perraudin[4,5], Mikael Ehn[1], Federico Bianchi[1]

[1] Institute for Atmospheric and Earth System Research / Physics, Faculty of Science, University of Helsinki, Finland

[2] Univ. Lyon, Université Claude Bernard Lyon 1, CNRS, IRCELYON, F-69626, Villeurbanne, France

[3] Aerodyne Research Inc., Billerica, Massachusetts 01821, USA

[4] Univ. Bordeaux, EPOC, UMR 5805, F-33405 Talence Cedex, France

[5] CNRS, EPOC, UMR 5805, F-33405 Talence Cedex, France

Correspondence: Haiyan Li (haiyan.li@helsinki.fi) and Matthieu Riva (matthieu.riva@ircelyon.univ-lyon1.fr)

**Abstract.** The capabilities of the recently developed Vocus proton-transfer-reaction time-of-flight mass spectrometer (PTR-TOF) are reported for the first time based on ambient measurements. With the deployment of the Vocus PTR-TOF, we present an overview of the observed gas-phase (oxygenated) molecules in the French Landes forest during summertime 2018 and gain insights into the atmospheric oxidation of terpenes, which are emitted in large quantities in the atmosphere and play important roles in secondary organic aerosol production. Due to the greatly improved detection efficiency compared to traditional PTR instruments, the Vocus PTR-TOF identifies a large amount of gas-phase signals with elemental composition categories including CH, CHO, CHN, CHS, CHON, CHOS, and others. Multiple hydrocarbons are detected, with carbon numbers up to 20. Particularly, we report the first direct observations of low-volatility diterpenes in the ambient air. The diurnal cycle of diterpenes is similar to that of monoterpenes and sesquiterpenes, but contrary to that of isoprene. Various types of terpene reaction products and intermediates are also characterized. Generally, the more oxidized products from terpene oxidations show a broad peak in the day due to the strong photochemical effects, while the less oxygenated products peak in the early morning and/or in the evening. To evaluate the importance of different formation pathways in terpene chemistry, the reaction rates of terpenes with main oxidants (i.e., hydroxyl radical, OH; ozone, $O_3$; and nitrate radical, $NO_3$) are calculated. For the less oxidized non-nitrate monoterpene oxidation products, their morning peaks likely have contributions from both $O_3$- and OH-initiated monoterpene oxidation. Due to the decreased OH concentration at night, monoterpene ozonolysis becomes more important in the evening. For the monoterpene-derived organic nitrates, oxidations by $O_3$, OH, and $NO_3$ radicals all contribute to their formation, with their relative roles varying considerably over the course of the day. Through a detailed analysis of terpene chemistry, this study demonstrates the capability of the Vocus PTR-TOF in the detection of a wide range of oxidized reaction products in ambient and remote conditions, which highlights its importance in investigating atmospheric oxidation processes.

## 1. Introduction

Organic aerosol (OA) constitutes a large fraction of atmospheric particles, having significant impacts on climate change, air quality, and human health (Maria et al., 2004; IPCC, 2013; Mauderly and Chow, 2008). On a global scale, secondary OA (SOA) is the largest source of OA, formed through the oxidation of volatile organic compounds (VOCs) (Jimenez et al., 2009). Biogenic VOCs (BVOCs) are released into the atmosphere in high amounts, with an annual global budget being 760 Tg C (Sindelarova et al., 2014). On average, SOA production from biogenic precursors ranges from 2.5 to 44.5 Tg C annually, which is much larger than that from anthropogenic sources (Tsigaridis and Kanakidou, 2003). Over the past few years, a considerable amount of studies have been conducted to investigate the atmospheric chemistry of BVOCs ( Calfapietra et al., 2013; Jokinen





et al., 2015; Ng et al., 2017). However, an incomplete understanding of BVOCs characteristics and their oxidation processes
in the atmosphere remains and yields large uncertainties in quantitative estimates of air quality and climate effects of
atmospheric aerosols (Carslaw et al., 2013; Zhu et al., 2019).

Terpenes make up the main fraction of BVOCs (Guenther et al., 1995), encompassing isoprene ($C_5H_8$), monoterpenes

($C_{10}H_{16}$), sesquiterpenes ($C_{15}H_{24}$), diterpenes ($C_{20}H_{42}$) and even larger compounds. With one or more C=C double bonds in
their molecular structures, terpenes are highly reactive. After entering the atmosphere, terpenes can undergo oxidative
chemistry with the common atmospheric oxidants including hydroxyl radical (OH), ozone ($O_3$), and nitrate radical ($NO_3$).
These oxidation processes generate a large variety of organic species, with volatilities ranging from gas-phase volatile species
(VOC), to semi-volatile / low volatility organic compounds (SVOC and LVOC), to extremely low volatility organic
compounds (ELVOC), which irremediably contribute to SOA formation (Donahue et al., 2012). Due to the chemical
complexity and low concentrations of BVOCs oxidation products, it remains extremely challenging to provide a
comprehensive understanding of terpene chemistry in the atmosphere.

With a high time response and sensitivity, proton-transfer-reaction mass spectrometry (PTR-MS) has been widely

used to study the emissions and chemical evolution of VOCs in the atmosphere (Yuan et al., 2017). However, due to
instrumental wall losses, previous PTR-MS instruments were not optimized to detect low volatility compounds. For example,
only a few ambient PTR-MS observations of sesquiterpenes are available (Kim et al., 2009; Jardine et al., 2011; Hellén et al.,
2018). Correspondingly, it is not surprising that ambient observations of diterpenes, which are generally considered to be non-
volatile compounds, have never been reported. In addition, the existing PTR-MS is often not sensitive enough to quantify
terpene oxidation products at atmospherically relevant concentrations (Yuan et al., 2017). To address these instrumental
limitations, two new versions of PTR were recently developed, the PTR3 (Breitenlechner et al., 2017) and the Vocus PTR-
TOF (Krechmer et al., 2018), both coupled with a time of flight (TOF) mass analyzer. With the drastically enhanced
sensitivities, these instruments are capable in detecting broader spectrum of VOCs, where the detection of low-volatility VOCs
is significantly improved compared to the traditional PTR-MS. Based on the laboratory evaluation by Riva et al. (2019a), the
Vocus PTR-TOF is able to measure both monoterpenes and lots of monoterpene oxidation products containing up to 6 oxygen
atoms.

Known for strong monoterpene emitters (Simon et al., 1994), the Landes forest in southwestern France is a suitable

place to investigate atmospheric terpene chemistry. A previous study at this site reported a high nocturnal monoterpene loading
and suggested that monoterpene oxidations play an important role in formation of new particles and the consequent growth of
atmospheric particles (Kammer et al., 2018). To better assess the roles of BVOCs in aerosol formation, the Characterization
of Emissions and Reactivity of Volatile Organic Compounds in the Landes Forest (CERVOLAND campaign) took place in
July 2018. The recently developed Vocus PTR-TOF was deployed in the CERVOLAND campaign to characterize terpenes
and their gas-phase oxidation products, which provides the first Vocus PTR-TOF measurement in a forested environment. In
this work, we present a comprehensive summary of the identified gas-phase molecules and gain insights into terpene chemistry
to demonstrate the Vocus PTR-TOF capabilities and the importance of its applications in atmospheric sciences.
Characterizations of isoprene, monoterpenes, sesquiterpenes, and particularly the rarely detected diterpenes, are reported. By
comparing the reaction rates of different formation pathways, we explore the formation mechanisms of terpene oxidation
products, including both non-nitrate and organic nitrate compounds.
**2. Experimental methods**
**2.1 Measurement site**
The Vocus PTR-TOF measurements were performed from 8 to 20 July, 2018 in the Landes forest (44º29'39.69"N,
0º57'21.75"W), as part of the CERVOLAND field campaign. The sampling site is situated at the European Integrated Carbon



Observation System (ICOS) station at Bilos in southwestern France along the Atlantic coast, ~40 km southwest from the
nearest urban area of the Bordeaux metropole. A detailed description of the site has been given by Moreaux et al. (2011).
Briefly, both population density and industrial emissions are low in this area. The forest is largely composed of maritime pines,
known as a strong monoterpene emitter (Simon et al., 1994), which provides a good place for BVOCs characterization.
**2.2 Instrumentation**
Compared to the traditional PTR instrument, the Vocus PTR-TOF used in this study is mainly differentiated in the following
aspects:

1.  a new chemical ionization source with a low-pressure reagent-ion source and focusing ion-molecule reactor (FIMR),

2.  no dependence of the sensitivity on ambient sample humidity due to the high water mixing ratio (10-20 % v/v) in the

FIMR,

3.  employment of a TOF mass analyzer with a longer flight tube and faster sampling data acquisition card (mass

resolving power up to 15 000 m/dm),

4.  an enhanced inlet and source design that minimizes contact between analyte molecules and inlet/source walls,

enabling detection of semi- and low-volatility compounds in a similar manner as chemical ionization mass

spectrometer (CIMS) instruments (Liu et al. 2019).


Details about the Vocus PTR-TOF are well described by Krechmer et al. (2018). Compared to the ionization in a traditional
PTR-MS at 2.0-4.0 mbar, a nitrate CIMS at ambient pressure, and an iodide CIMS at around 100 mbar, we operate the Vocus
ionization source at a low pressure of 1.0-1.5 mbar. During the campaign, the Vocus PTR-TOF measurements were performed
at around 2 m above ground level (a.g.l). Sample air was drawn in through 1-m long PTFE tubing (10 mm o.d., 8 mm i.d.)
with a flow rate of 4.5 L min$^{-1}$, which helped to reduce inlet wall losses and sampling delay. Of the total sample flow, only
150 sccm went into the Vocus, while the remainder was directed to the exhaust. The design of FIMR consists of a glass tube
with a resistive coating on the inside surface and four quadrupole rods mounted radially on the outside. With an RF field, ions
are collimated to the central axis, improving the detection efficiency of product ions. The mass resolving power of the 1.2 m
long TOF mass analyzer was 12 000-13 000 m/dm during the whole campaign. Data were recorded with a time resolution of
5 s. Background measurements using high purity nitrogen (UHP $N_2$) were automatically performed every hour.

The temperature, relative humidity (RH), wind speed, and ambient pressure were continuously monitored at 3.4 m

a.g.l. whereas the solar radiation was measured at 15.6 m a.g.l from a mast located at the site. The mixing ratios of nitrogen
oxides ($NO_x$) and ozone ($O_3$) were measured at 4 m a.g.l with UV absorption and chemiluminescence analyzers, respectively.
All data are reported in Coordinated Universal Time (UTC).
**2.3 Data analysis and quantification of multiple compounds**
Data analysis was performed using the software package "Tofware" (https://www.tofwerk.com/software/tofware/) that runs in
the Igor Pro environment (WaveMetrics, OR, USA). Tofware enables the time-dependent mass calibration, baseline
subtraction, and assignment of a molecular formula to the identified ions by high resolution analysis. Signals were averaged
over 30 min before mass calibration. Due to the high resolving power of the LTOF mass analyzer, isobaric ions were more
clearly separated. Examples of peak identification are given in Fig. S1.

The Vocus was calibrated twice a day during the campaign with a mixture (70 ppb each) of terpenes (*m/z* 137:

alpha/beta pinene + limonene; *m/z* 135: cymene) that was diluted using UHP $N_2$. Similar to traditional PTR instrument, the
sensitivities for different VOCs in the Vocus are linearly related to their rate constants of the proton-transfer reactions
(Cappellin et al., 2012; Krechmer et al., 2018). Using the calculated sensitivities of monoterpenes and cymene from calibration
data and their respective rate constants (*k*), an empirical relationship between the sensitivity and *k* was built from the





scatterplots using linear regression: Sensitivity $= 509.75 \times k$. Once $k$ is available, the sensitivity of a compound can be predicted.
It should be noted that both monoterpenes and cymene fragment inside the instrument. The predicted sensitivities with this
method may be underestimated for compounds which do not fragment or fragment less than monoterpenes and cymene inside
the PTR instruments. Rate constants for the proton-transfer reactions have only been measured for a subset of compounds. To
quantify terpenes and their oxidation products, we used the method proposed by Sekimoto et al. (2017) to calculate the rate
constants of different compounds with the polarizability and permanent dipole moment of the molecule. According to
Sekimoto et al. (2017), the polarizability and dipole moment of a molecule can be obtained based on the molecular mass,
elemental composition, and functionality of the compound. For a class of VOCs with the same number of electronegative
atoms, their polarizabilities can be well described using their molecular mass (Sekimoto et al., 2017). For VOCs containing a
specific functional group, it is found that their dipole moments are relatively constant based on results in the CRC Handbook
(Lide, 2005). Since no isomer information is provided by mass spectrometry alone, it is challenging to figure out the
functionality of different compounds. Therefore, the polarizability and dipole moment of the compounds observed in this study
were estimated only based on the molecular mass and elemental composition. In this work, based on the physical properties
of various compounds in CRC Handbook (Lide, 2005) and the results in Sekimoto et al. (2017), we built the functions between
polarizability ($\alpha$) and molecular mass ($M_R$) for different groups of VOCs and calculated the average dipole moment ($\mu$) for
each group. For example, the polarizabilities of hydrocarbons were approximated as $\alpha = 0.142\,M_R - 0.3$ and the dipole moment
was approximated to be zero. For the non-nitrate oxygenated compounds with one oxygen, $\alpha = 0.133\,M_R - 1.2$, and the dipole
moment was averaged to be 1.6.
It should be noted that uncertainties are introduced to the calculated sensitivities in the following factors. First,
oxidized compounds usually fragment more than terpene precursors in PTR instruments. For instance, alcohol-containing
compounds easily split off water and undergo the highest degree of fragmentation (Buhr et al., 2002). A study by Kari et al.
(2018) showed that around 95.5% of 1,8 – cineole fragmented with an reduced electric field (E/N) of 130 Td. Molecules
containing other functional groups fragment to varying, but lesser degrees; however, the theoretically calculated sensitivities
here should be regarded as upper limits for terpene oxidation products. Further, some low-volatility compounds may
experience wall losses to varying extents inside the inlet tubing and the instrument and therefore have worse transmissions.
The method in this work may overestimate the sensitivities of these low-volatility compounds. In addition to proton transfer
reactions, some VOCs can be ionized through ligand switching reactions with water cluster $((H_2O)_nH_3O^+)$ (Tani et al., 2004),
thus increasing their sensitivity. However, with the calibration standards used in this study, it is hard to estimate the effect of
ligand switching ionization. Lastly, uncertainties come from the estimation of polarizability and dipole moment of a molecule.
With the method used in this study, the sensitivity is calculated to be within 50% error when only the elemental composition
of a compound is known (Sekimoto et al., 2017).
**3. Results and discussion**
**3.1 Meteorology and trace gases**
Figure 1 displays the time variations of meteorological conditions and trace gases during the observation period. The weather
was mostly sunny, with solar radiation varying from 400 to 800 W/m$^2$ during daytime, indicating strong photochemical activity.
The ambient temperature and RH varied regularly every day. On average, the temperature was $22.8 \pm 5.9\ °C$, ranging from
12.1 to 35.0 °C, which is favorable for BVOCs emissions in the forest. The average RH was $70.5 \pm 19.0\ \%$ during the campaign.
Generally, the air masses were quite stable within the canopy. The wind speed never exceeded 1 m/s, indicating the major
influence of local sources on atmospheric processes in this study.
The $O_3$ levels fluctuated dramatically between day and night during the campaign. The average $O_3$ diurnal cycle
showed that $O_3$ concentration peaked up to ~50 ppb in the daytime. However, during most of the nights, $O_3$ concentration





dropped below 2 ppb. Considering the high nighttime concentration of terpenes observed by the previous study at this site in
the same season (Kammer et al., 2018), the low $O_3$ level at night suggests the full consumption of $O_3$ by terpenes. Such
reactions of terpenes with $O_3$ can produce low volatility organic compounds, thus contributing to SOA formation (Presto et al.,
2005; Jokinen et al., 2014).

The NO concentration was generally low during the campaign, below detection limit (i.e., <0.5 ppb) most of the time.

However, clear NO plumes was sometimes observed in the early morning, as shown in Fig.1e. The NO concentration peak at
4 am is probably the combination of local emission sources and low boundary layer. With the increasing sunlight afterwards,
the NO concentration started to decrease. A similar diel pattern of $NO_2$ was observed by the previous study at this site (Kammer
et al., 2018). The lower $NO_2$ concentration during daytime is likely explained by dilution with increasing boundary layer height
and $NO_2$ photolysis.
**3.2 Vocus PTR-TOF capabilities in the forest**
While Krechmer et al. (2018) and Riva et al. (2019a) have described the novel setup and performance of the Vocus PTR-TOF
and its application during a lab study, the instrument capability has not been fully explored in an ambient environment. Based
on the CERVOLAND deployment, we provide here, the first overview of gas-phase molecules measured by the Vocus PTR-
TOF in the forest. For a better visualization of the complex data set from real atmosphere, mass defect plots (averaged over
the whole campaign) are shown in Fig. 2 with the difference between the accurate mass and the nominal mass of a compound
plotted against its accurate mass. With the addition of hydrogen atoms, the mass defect increases, while the addition of oxygen
atoms decreases the mass defect. Therefore, changes in the mass defect plot help to provide information on chemical
transformation such as oxidation.

The mass defect plot in Fig. 2a is colored according to the retrieved elemental composition, with the black circle

indicating unidentified molecules. The size of the markers is proportional to the logarithm of the peak area of the molecule.
During the campaign, the Vocus PTR-TOF detected large amounts of (O)VOCs, with elemental composition categories of CH,
CHO, CHN, CHS, CHON, CHOS, and others. For hydrocarbons, multiple series with different carbon numbers were measured,
with compounds containing 5 carbon atoms ("$C_5$"), 10 carbon atoms ("$C_{10}$"), 15 carbon atoms ("$C_{15}$"), and 20 carbon atoms
("$C_{20}$") highlighted in the figure. Compared to the traditional PTR instruments, the observation of larger hydrocarbon
molecules by the Vocus PTR-TOF is mainly caused by the much lower wall losses and increased detection efficiency.
Hydrocarbon signals were largely contributed by monoterpene ($C_{10}H_{16}H^+$) and its major fragment ($C_6H_8H^+$), indicating the
monoterpene-dominated environment in the Landes forest (Kammer et al., 2018). According to previous studies, monoterpene
emissions in the Landes forest are dominated by α-pinene and β-pinene (Simon et al., 1994; Kammer et al., 2018). The
identified compound with the elemental composition of $C_4H_9^+$ ranked the third largest peak in hydrocarbons. One possible
explanation for $C_4H_9^+$ peak could be the protonated butene, which is emitted by vegetations or from anthropogenic sources
(Goldstein et al., 1996; Zhu et al., 2017). The fragmentation of butanol also produces $C_4H_9^+$ signal. Like many other alcohols,
butanol can easily lose an OH during ionization in PTR sources (Spanel and Smith, 1997). During the measurements at the
Station for Measuring Ecosystem-Atmosphere Relations (SMEAR II) site in Hyytiälä, Finland, Schallhart et al. (2018)
concluded that $C_4H_9^+$ signal detected by PTR-TOF mainly came from butanol used by aerosol instruments, i.e., condensation
particle counters (CPCs). In this study, CPCs using butanol to measure the particle concentration were also deployed at the
site. While the exhaust air emitted from these collocated instruments was filtered using charcoal denuder, we cannot exclude
the contribution of butanol to the identified $C_4H_9^+$ signal. The spiky peaks in the time series of $C_4H_9^+$ compound also indicated
the influence of butanol (Fig. S2). Finally, the green leaf volatiles (GLV), a group of six-carbon aldehyde, alcohols and their
esters which can be directly released by the plants, have been found to fragment at $m/z$ 57 inside the PTR instruments (Rinne
et al., 2005; Pang, 2015) and may also contribute to the observed $C_4H_9^+$ signal.





In addition to the emitted precursors, the Vocus PTR-TOF detected various VOCs reaction products and
intermediates. Similar to the PTR3 measurements in the CLOUD chamber (Breitenlechner et al., 2017), many oxygenated
compounds from terpene reactions with varying degrees of oxidation were observed in this study. However, as a potential
limitation of the instrument, no dimers in the atmosphere were identified by the Vocus PTR-TOF, consistent with the results
from a previous laboratory deployment (Riva et al., 2019a). Several cyclic volatile methyl siloxanes (VMS) were measured,
which have been recently reported by traditional PTR-TOF instruments (Yuan et al., 2017). Cyclic VMS are silicon-containing
compounds widely used in cosmetics and personal care products (Buser et al., 2013; Yucuis et al., 2013). In this study, the
identified peaks of cyclic VMS were protonated D3 siloxane ($C_6H_{18}O_3Si_3$), D4 siloxane ($C_8H_{24}O_4Si_4$), D5 siloxane
($C_{10}H_{30}O_5Si_5$), D6 siloxane ($C_{12}H_{36}O_6Si_6$), and their $H_3O^+$ cluster ions. The existence of these peaks in the mass spectra helps
to extend the range in mass or peak-width calibrations.
Figure 2b compares the daytime and nighttime variations of different molecules, with the marker sized by the signal
difference between day and night. The daytime periods cover from 4:30 am to 7:30 pm, and the nighttime periods are from
7:30 pm to 4:30 am of the next day (both are UTC time). The data points are colored in orange when the nighttime signal of
the compound is larger than its daytime signal, and in green when the daytime signal is higher. Patterns in the figure clearly
show the difference in the diurnal variations of gas molecules with different oxidation degrees. For example, most
hydrocarbons are characterized with higher concentrations at night, which is largely caused by the stable nocturnal boundary
layer. The more oxidized compounds with more oxygen numbers are generally more abundant during the day due to enhanced
photochemistry, whereas the concentrations of the less oxidized compounds are mostly higher at night. Details on the diurnal
profiles of different oxidation products and their formation mechanisms are provided in Sect. 3.4.
**3.3 Terpene characteristics**
The characterizations of isoprene, monoterpenes, sesquiterpenes, and the rarely reported diterpenes, are investigated in this
study (Fig. 3, Fig. 4). On the global scale, isoprene is the most emitted BVOC species. It has been well established that
photooxidation of isoprene in the atmosphere contributes to SOA formation through the multiphase reactions of isoprene-
derived oxidation products (Claeys et al., 2004; Henze and Seinfeld, 2006; Surratt et al., 2010). However, recent advances on
isoprene chemistry found that isoprene can impact both particle number and mass of monoterpene-derived SOA by scavenging
hydroxyl and peroxy radicals (Kiendler-Scharr et al., 2009; Kanawade et al., 2011; McFiggans et al., 2019). During the
CERVOLAND campaign, the average mixing ratio of isoprene was 0.6 ppb, consistent with the mean value of 0.4 ppb reported
for the LANDEX campaign during summer 2017 at the same site (Mermet et al., 2019). These values are much lower than that
in the southeastern United States (Xiong et al., 2015) and Amazon rainforest (Wei et al., 2018) but higher than observations
in the boreal forest at the SMEAR II station (Hellén et al., 2018). Isoprene emissions require sunlight (Monson et al., 1989).
Therefore, a pronounced diurnal pattern of isoprene was observed with maximum mixing ratios occurring during daytime and
minima at night.
As expected, monoterpenes showed the highest mixing ratios among all the terpenes, with an average value of 6.0
ppb. On July 9, a heavy monoterpene episode occurred at night, with the monoterpene mixing ratio reaching as high as 41.2
ppb. Comparatively, the average monoterpene level observed in this work is similar to the measurements performed in 2015
and 2017 at the same site (Kammer et al., 2018; Mermet et al., 2019) and more than ten times higher than that observed in the
boreal forest at SMEAR II in summer (Hakola et al., 2012; Hellén et al., 2018). The high concentration of monoterpenes
indicates the potential significance of monoterpene-related aerosol chemistry in the Landes forest. Opposite to the diurnal
variations of isoprene, monoterpene concentrations peaked at night, caused by the stable nocturnal boundary layer. During
daytime, the concentration of monoterpenes dropped to around 0.9 ppb, due to the increased atmospheric mixing after sunrise
and the rapid photochemical consumptions.



A study in Hyytiälä concluded that sesquiterpenes, due to their higher reactivity, could play a more important role in
$O_3$ chemistry than monoterpenes, even though the concentration of sesquiterpenes was much lower (Hellén et al., 2018).
However, the short lifetimes of sesquiterpenes also mean that their concentrations will be highly dependent on the sampling
location at a given site. Some studies also proposed that sesquiterpene oxidation products are linked to atmospheric new particle
formation (Bonn and Moortgat, 2003; Boy et al., 2007). Despite the potential importance of sesquiterpenes in aerosol chemistry,
the available data on ambient sesquiterpene quantification remains still quite limited. In this work, the mixing ratios of
sesquiterpenes were found to vary from 8.9 ppt to 408.9 ppt in the Landes forest, with an average of 64.5 ppt during the
observations. This sesquiterpene level is comparable to that reported by Mermet et al. (2019) in summer 2017 at the same site
and observations by Jardine et al. (2011) in Amazonia but higher than previous measurements at SMEAR II station (Hellén et
al., 2018). As shown in Fig. 4, sesquiterpenes displayed a similar diurnal pattern with monoterpenes, consistent with
observations in other areas (Jardine et al., 2011; Hellén et al., 2018).
While diterpenes are present in all plants in the form of phytol, they have been thought for a long time not to be
released by vegetation due to their low volatility (Keeling and Bohlmann, 2006). In 2004, von Schwartzenberg et al. (2004)
reported for the first time the release of plant-derived diterpenes into the air. A recent study found that the emission rate of
diterpenes by Mediterranean vegetation was in the same order of magnitude as monoterpenes and sesquiterpenes (Yáñez-
Serrano et al., 2018). For the first time, this study reports the ambient concentration of diterpenes in a forest. According to the
Vocus PTR-TOF measurements, the average mixing ratio of diterpenes was around 2 ppt in the Landes forest. Considering the
low volatility of diterpenes and their potential wall losses inside the inlet tubing and the instrument, the diterpene concentration
might be higher. Similar to monoterpenes and sesquiterpenes, diterpenes presented peak concentrations at night and lower
levels during the day. Although the amounts of diterpenes in the atmosphere are hundreds to thousands times lower than those
of monoterpenes and sesquiterpenes, diterpenes potentially play a role in atmospheric chemistry due to their unsaturated
structure and high molecular weight (Matsunaga et al., 2012). Up to now, there is no report on the possible atmospheric
implications of diterpenes, which should deserve more attention in the future.
Considering the similar atmospheric behaviors of monoterpenes, sesquiterpenes, and diterpenes in this study, it is
questioned if the observed sesquiterpenes and diterpenes are real signals in the atmosphere or generated by monoterpenes in
the instrument. Figure 5 illustrates the scatter plots among monoterpenes, sesquiterpenes, and diterpenes, colored by time of
the day. At night, both sesquiterpenes and diterpenes correlated well with monoterpenes. However, their correlation with
monoterpenes got weaker during daytime as the data points became more scattered. This suggests that the observations of
sesquiterpenes and diterpenes are real emissions in the atmosphere. Comparatively, sesquiterpenes and diterpenes showed a
strong correlation with each other through the whole day ($r^2$ = 0.85).
**3.4 Insights into terpene chemistry**
**3.4.1 Comparison with chamber results**
Due to the diverse precursors and changing environmental conditions in the ambient air, it is challenging to retrieve all the
atmospheric chemical processes occurring within the Landes forest. To start with, we compare the ambient data with those
from α-pinene ozonolysis in the presence of $NO_x$ conducted in the COALA chamber at the University of Helsinki. A detailed
description of the laboratory experiment is provided elsewhere (Riva et al., 2019a, 2019b). According to literature,
monoterpenes undergo some degree of fragmentation within the PTR instrument, producing dominant ions of $C_6H_9^+$, $C_5H_7^+$,
$C_7H_{11}^+$, et al (Tani et al., 2003, 2013; Kari et al., 2018). As illustrated in Fig. 6, $C_6H_9^+$ is the largest fragment produced by
monoterpenes within the Vocus PTR-TOF. However, a clear difference of monoterpene fragmentation pattern is observed in
the mass spectra of ambient observations and chamber experiments. While the signal of $C_6H_9^+$ is lower than that of $C_{10}H_{17}^+$
during the field deployment, $C_6H_9^+$ peak is higher than $C_{10}H_{17}^+$ peak in the chamber study. Based on the monoterpene
calibration data, the $C_6H_9^+$ signal is around 40% and 138% of the protonated monoterpene signal in ambient deployment and





chamber experiment, respectively. The larger presence of the $C_6H_9^+$ peak in the chamber study can be likely explained by the
much higher concentrations of oxygenated terpenoids during the chamber experiments. Indeed, previous studies have shown
that oxygenated terpenoids, including linalool and pinonaldehyde, fragment inside the PTR instrument and produce a dominant
ion at *m/z* 81 (Maleknia et al., 2007; Tani, 2013). Different settings of the instrument in the two studies can also contribute to
the difference, i.e., the Vocus pressure, the drift voltage, and different mass transmission functions of the instrument (Tani et
al., 2003, 2013; Kari et al., 2018). In addition, the fragmentation patterns vary among individual monoterpene species due to
their different physicochemical properties (Tani et al., 2013; Kari et al., 2018). Considering that α-pinene is the only
monoterpene species injected in the chamber experiment, the combination of various monoterpenes in the atmosphere likely
introduces additional differences in the fragmentation pattern.

Gas-phase ozonolysis of alkenes generates OH radicals in high yields (Rickard et al., 1999). Without an OH scavenger,

both $O_3$- and OH-initiated oxidations happened during α-pinene ozonolysis in the chamber. Using the Vocus PTR-TOF,
various oxidation products were identified in the chamber study, with the dominant species being $C_7H_{10,12}O_{3-6}$, $C_8H_{14}O_{3-6}$,
$C_9H_{14}O_{1-5}$, and $C_{10}H_{14,16}O_{2-6}$. In comparison, more oxygenated compounds which were directly emitted or from monoterpene
reactions were observed in ambient air due to complex environmental conditions, with the oxygen number ranging from 1 to
7. Therefore, the Vocus PTR-TOF measurements provide the opportunity to characterize both the emitted precursors and the
resulting oxidation products. During the chamber experiments, $NO_2$ was injected and photolyzed using 400nm LED lights to
generate NO. In the presence of $NO_x$, organic nitrates were formed from the reactions between NO and monoterpene-derived
peroxy radicals ($RO_2$). The major organic nitrates observed were $C_9H_{13,15}NO_{6-8}$ and $C_{10}H_{13,15}NO_{3-8}$. Compared to the chamber
study, more organic nitrates of $C_8$, $C_9$, and $C_{10}$ from monoterpene reactions were identified in CERVOLAND data. It is worth
pointing out that the combination of different monoterpene species in the ambient environment may result in various types of
organic nitrates through different formation pathways.
**3.4.2 Non-nitrate terpene oxidation products**
Based on the ambient observations, the non-nitrate oxidation products from isoprene, monoterpenes, and sesquiterpenes, are
investigated in this study. Isoprene gas-phase products are mainly represented by $C_4$ and $C_5$ compounds (Wennberg et al.,
2018). In this work, we consider $C_4H_{6,8}O_n$ and $C_5H_{8,10,12}O_n$ (n=1~6) as the dominant non-nitrate products from isoprene
oxidations. The diurnal variations of $C_5H_8O_n$ are displayed in Fig. 7 and the others in Fig. S3-5. Generally, all these oxidation
products displayed an evening peak at around 8 pm, which may come from the $O_3$- or OH-initiated isoprene oxidations.
Globally, reactions with $O_3$ contribute a small fraction of approximately 10% to isoprene removal in the atmosphere (Wennberg
et al., 2018). When isoprene reacts with $O_3$, one carbon is always split off from the molecule (Criegee, 1975). Considering the
peak concentration of isoprene at 8 pm and the relatively high $O_3$ concentration at the moment (Figs. 1 and 4), isoprene
ozonolysis is likely contributing to the formation of $C_4$ oxidation products. Because OH radicals can be efficiently produced
from alkene ozonolysis (Pfeiffer et al., 2001), the OH-initiated oxidation of isoprene can also be an important formation
pathway of these oxidation products in the evening. For example, as a predominant product from the reactions of isoprene with
OH, $C_5H_{10}O_3$ (corresponding to isoprene hydroxy hydroperoxide and/or isoprene epoxydiols) presented a clear single peak in
the evening. To determine the relative importance of $O_3$- and OH-initiated oxidations in isoprene chemistry at night, the
reaction rate (R) of isoprene with $O_3$ and OH radical were compared by Eq. (1) and Eq. (2):
$R_{ISO+OH} = k_{ISO+OH}[ISO][OH]$       (1)
$R_{ISO+O3} = k_{ISO+O3}[ISO][O_3]$       (2)
where *k* is the reaction rate coefficient of isoprene with OH or $O_3$, and [ISO], [OH] or $[O_3]$ is the concentration of isoprene,
OH radical or $O_3$.
Taking the evening peak of isoprene oxidation products at 8 pm as an example, we compared the roles of $O_3$ and OH radicals
in their formation. Laboratory studies have shown that the reaction rate coefficient of isoprene with OH radical is generally





$10^7$ times larger than that of isoprene with O$_3$ (Dreyfus et al., 2002; Kari et al., 2004). According to literature, the nighttime
concentration of tropospheric OH radical varies in the range of $1 \times 10^4 - 1 \times 10^5$ molecules cm$^{-3}$ ($0.0004 - 0.004$ ppt) in the
field (Shirinzadeh et al., 1987; Khan et al., 2008; Petäjä et al., 2009; Stone et al., 2012). Therefore, with an O$_3$ concentration
of ~20 ppb at 8 pm, if the OH concentration was around $1 \times 10^4$ molecule cm$^{-3}$ (0.0004 ppt) at the moment, the reaction rate
of isoprene with OH radical was around 0.2 times as high as that of isoprene with O$_3$. If the OH concentration reached up to 1
$\times 10^5$ molecule cm$^{-3}$ (0.004 ppt) at 8 pm, the reaction rate of isoprene with OH radical was 2 times higher than that of isoprene
with O$_3$. For the more oxidized compounds from isoprene oxidations, their concentrations had a broad daytime presence from
10 am to 8 pm due to strong photooxidation processes. Similar diurnal variations of C$_4$H$_{6,8}$O$_{5,6}$ and C$_5$H$_{8,10,12}$O$_{5,6}$ measured by
nitrate CIMS have been observed in an isoprene-dominated environment at Centreville, Alabama (Massoli et al., 2018).

The diurnal patterns of C$_8$H$_{12,14}$O$_n$, C$_9$H$_{14}$O$_n$, and C$_{10}$H$_{14,16,18}$O$_n$ (n=1~6) were illustrated to characterize monoterpene

oxidations in the Landes forest (Fig. 8; Fig. S6-10). For the less oxidized compounds with oxygen numbers from 1 to 4, most
of them were observed with clear morning and evening peaks, which can be produced from O$_3$- and OH-initiated monoterpene
oxidations. For the morning peak at around 7 am, the relative roles of O$_3$- and OH-initiated monoterpene oxidation were
evaluated using the similar method as in Eq. (1) and Eq. (2). The reaction rate coefficient of monoterpene + OH is
approximately $10^6$ times higher than that of monoterpene + O$_3$ (Atkinson et al., 1990; Khamaganov and Hites, 2001; Gill and
Hites, 2002; Hakola et al., 2012). In the morning, typical tropospheric OH concentrations have been observed to be around 1
$\times 10^5 - 1 \times 10^6$ molecule cm$^{-3}$ ($0.004 - 0.04$ ppt) (Shirinzadeh et al., 1987; Ren et al., 2003; Khan et al., 2008; Petäjä et al.,
2009; Stone et al., 2012). For an OH concentration of $1 \times 10^5$ molecule cm$^{-3}$ (0.004 ppt), with the average O$_3$ concentration of
15 ppb at 7 am, the reaction rate of monoterpene + OH was about 0.25 times as high as that of monoterpene + O$_3$. If the OH
concentration was up to $1 \times 10^6$ molecule cm$^{-3}$ (0.04 ppt) at 7 am, the reaction rate of monoterpene with OH radical was 2.5
times higher than that of monoterpene with O$_3$ according to the calculations. In other words, both oxidants are likely to be of
importance at this time. For the evening peak of the less oxidized monoterpene oxidation products at 8 pm, the relative
importance of O$_3$ and OH radical in monoterpene chemistry changed due to the lower OH concentration. With the average O$_3$
concentration of ~20 ppb at 8 pm, a similar analysis as above resulted in O$_3$ reactions being 5-50 times more important than
OH radical reactions with monoterpenes, indicating that the evening peaks are mainly from ozonolysis. Compared to other
compounds, the evening peak of C$_9$H$_{14}$O, C$_{10}$H$_{16}$O, C$_{10}$H$_{18}$O, and C$_{10}$H$_{18}$O$_2$ extended over midnight. C$_9$H$_{14}$O has been found
to be one of the main products formed in the ozonolysis reactions of monoterpenes (Atkinson and Arey, 2003). O$_3$-initiated
oxidation with extremely high monoterpene levels might be responsible for the high concentration of C$_9$H$_{14}$O at night.
Camphor (C$_{10}$H$_{16}$O), linalool (C$_{10}$H$_{18}$O), and linalool oxide (C$_{10}$H$_{18}$O$_2$) can be emitted by leaves and flowers (Corchnoy et al.,
1992; Lavy et al., 2002). Therefore, direct emissions from vegetation in the Landes forest may contribute to the high mixing
ratios of these compounds during night. With strong photochemical oxidations during the day, the diurnal cycles of the more
oxidized compounds were characterized with a broad daytime distribution peaking between 2:00 pm and 4:00 pm UTC.

To date the oxidation processes of sesquiterpenes have been rarely investigated despite its potential significance in

new particle formation and SOA formation (Bonn and Moortgat, 2003; Winterhalter et al., 2009). In this study, various
sesquiterpene oxidation products were observed, mainly including C$_{14}$H$_{22}$O$_n$, C$_{15}$H$_{22}$O$_n$, and C$_{15}$H$_{24}$O$_n$ (n=1~6), providing the
possibility to explore the oxidations of sesquiterpenes in the atmosphere. As shown in Fig. 9 and Fig. S11-12, with the increase
of oxygen numbers, sesquiterpene oxidation products displayed similar variations in their diurnal profiles with monoterpene
oxidation products. The less oxidized products with 1 to 3 oxygen peaked both in the morning and in the evening, and the
more oxidized compounds had a broad presence throughout the day. These results indicate the similar oxidation processes of
sesquiterpenes with monoterpenes in the Landes forest.





### 3.4.3 Terpene-derived organic nitrates

Organic nitrates have been shown to represent a large fraction of submicron aerosol nitrate at both urban and rural sites in Europe (Kiendler-Scharr et al., 2016). During daytime, the reaction of peroxy radicals with NO can lead to the formation of organic nitrates. At night, $NO_3$ radicals from the oxidation of $NO_2$ by $O_3$, can also react with unsaturated compounds mostly coming from BVOCs to generate organic nitrates (Ayres et al., 2015). In this study, the less oxidized organic nitrates from monoterpene oxidations presented a distinct morning peak at 7 am (Fig. 11; Fig. S15-16), which can come from $O_3$- and OH-initiated monoterpene oxidations in the presence of $NO_x$. In addition, both isoprene- and monoterpene-derived organic nitrates showed evening peaks at around 8 pm (Fig. 10, Fig. S13-14). Using monoterpenes as an example, the relative roles of $O_3$, OH radical, and $NO_3$ radical in the nighttime formation of monoterpene-derived organic nitrates were evaluated by calculating the corresponding reaction rate (R):

$$R_{MT+O3} = k_{MT+O3}[MT][O_3] \tag{3}$$
$$R_{MT+OH} = k_{MT+OH}[MT][OH] \tag{4}$$
$$R_{MT+NO3} = k_{MT+NO3}[MT][NO_3] \tag{5}$$

where $k$ is the reaction rate coefficient of monoterpenes with $O_3$, OH radical or $NO_3$ radical, and [MT], [$O_3$], [OH] or [$NO_3$] is the concentration of monoterpenes, $O_3$, OH radical or $NO_3$ radical.

Taking the peak concentration of monoterpene-derived organic nitrates at 8 pm as an example, the concentration of $NO_3$ radical was calculated by assuming a steady state between its production from $O_3$ and $NO_2$ and its removal by oxidation reactions and losses. The details have been described by Allan et al. (2000) and Peräkylä et al. (2014). With the high $O_3$ scavenging by monoterpenes in the evening, the estimated concentration of $NO_3$ radical was 0.017 ppt. Using $k_{MT+O3} = 6.9\times10^{-17}$ cm$^3$ molecule$^{-1}$ s$^{-1}$ and $k_{MT+NO3} = 7.5\times10^{-12}$ cm$^3$ molecule$^{-1}$ s$^{-1}$ taken from Peräkylä et al. (2014), the reaction rate of monoterpenes with $O_3$ was ~10 times higher than that of monoterpenes with $NO_3$ radicals. However, while ozonolysis was likely to dominate the overall oxidation of monoterpenes, the organic nitrate formation from $O_3$-initiated oxidation may still be much lower than those from $NO_3$-initiated oxidations, depending on what fraction of $RO_2$ radicals were reacting with $NO_x$. The relative importance of $O_3$ and OH radical in monoterpene chemistry at this time was the same as discussed in Sect. 3.4.2.

### 4. Conclusions

This work presented the deployment of the new state-of-the-art Vocus PTR-TOF in the French Landes forest during the CERVOLAND campaign. The Vocus PTR-TOF capabilities are evaluated for the first time in the actual ambient environment by the identification of the observed gas-phase molecules. With the improved detection efficiency and measurement precision compared to the traditional PTR instruments, multiple hydrocarbons with carbon numbers varying from 3 to 20 were observed as well as various VOCs oxidation products. Hydrocarbon signals were dominated by monoterpenes and their major fragment ions (e.g., $C_6H_8H^+$) within the instrument, consistent with high monoterpene emissions in the Landes forest. In general, most hydrocarbon molecules and the less oxidized compounds were characterized with high signals at night, whereas the more oxidized compounds exhibited elevated intensity during the day.

To demonstrate the importance of Vocus PTR-TOF application in atmospheric science study, the characteristics of terpenes and their oxidation products were investigated. In addition to the observation of isoprene, monoterpenes, and sesquiterpenes, this study presented the ambient characteristics of the rarely recorded diterpenes, which are traditionally considered as non-volatile species in the atmosphere. On average, the concentration of diterpenes was 1.7 ppt in the Landes forest, which was hundred to thousand times lower than that of monoterpenes (6.0 ppb) and sesquiterpenes (64.5 ppt). However, considering their low vapor pressure and high reactivity, diterpenes may potentially play an important part in atmospheric chemistry. The diurnal variations of diterpenes showed the maximum peak at night and low levels during the day, similar to those of monoterpenes and sesquiterpenes.





With strong photochemical oxidations of terpenes during the day, the more oxidized terpene reaction products were

observed with a broad daytime peak, whereas the less oxidized terpene reaction products showed peak concentrations in the
early morning or/and in the evening. By calculating the reaction rates of terpenes with the main oxidants, OH radical, $O_3$, and
$NO_3$ radical, the contributions of different formation pathways to terpene oxidations were evaluated. The morning peaks of
non-nitrate terpene reaction products were contributed by both $O_3$- and OH-induced terpene oxidations. For the evening peaks
of non-nitrate terpene oxidation products, terpene ozonolysis played an increasing role due to the lower OH concentration at
night. For the formation of terpene-derived organic nitrates, the relative importance of $O_3$-, OH-, and $NO_3$- driven oxidation
pathways were more difficult to evaluate. Overall, we have shown that the Vocus PTR-TOF is able to detect a very broad
coverage of compounds, from VOCs precursors to various oxidation products. Therefore, the application of the Vocus PTR-
TOF in atmospheric sciences will be fundamental in understanding the chemical evolution of VOCs in the atmosphere and
their roles in air quality and climate issues.

**Author contributions**

ME and MR conceived the study. MR, LH, PF, EV, and EP conducted the field measurements. HL carried out the data analysis.
MR, PR, KD, JK, DW, MK, ME, and FB participated the data analysis. HL wrote the paper with inputs from all coauthors.

**Competing interests**

The authors declare that they have no conflict of interest.

**Acknowledgements**

This work was supported by the European Research Council under grants 742206 ATM-GP and 638703 COALA, and the
Academy of Finland (project numbers 317380 and 320094). The authors would like to thank the PRIMEQUAL programme
for financial support (ADEME, convention #1662C0024). This study has also been carried out with financial support from the
French National Research Agency (ANR) in the frame of the Investments for the future Programme, within the Cluster of
Excellence COTE (ANR-10-LABX-45) of the University of Bordeaux. Special thanks to Dr Elena Ormeño-Lafuente (IMBE)
for the loan of the BVOC calibration gas cylinders and Dr Christophe Chipeaux and Dr Denis Loustau (ISPA-INRA) for their
precious help in providing meteorological data and access to ICOS station facility.

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

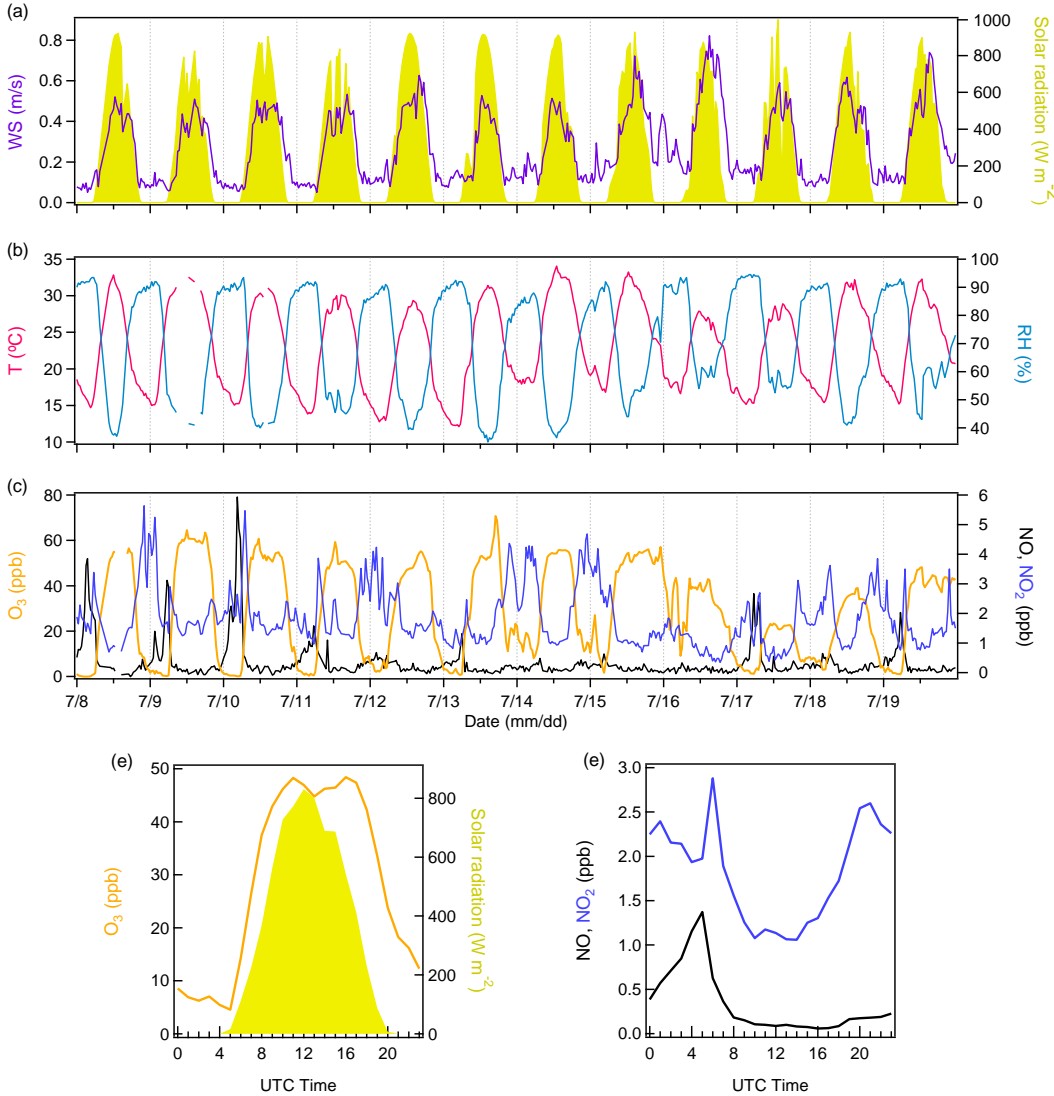

**753**

**Figure 1. Variations of meteorological conditions and trace gases. (a) Time series of wind speed and solar radiation. (b)**
**Time series of temperature and relative humidity. (c) Time series of O₃, NO, and NO₂. (d) Diurnal cycles of O₃ and**
**solar radiation. (e) Diurnal cycles of NO and NO₂.**

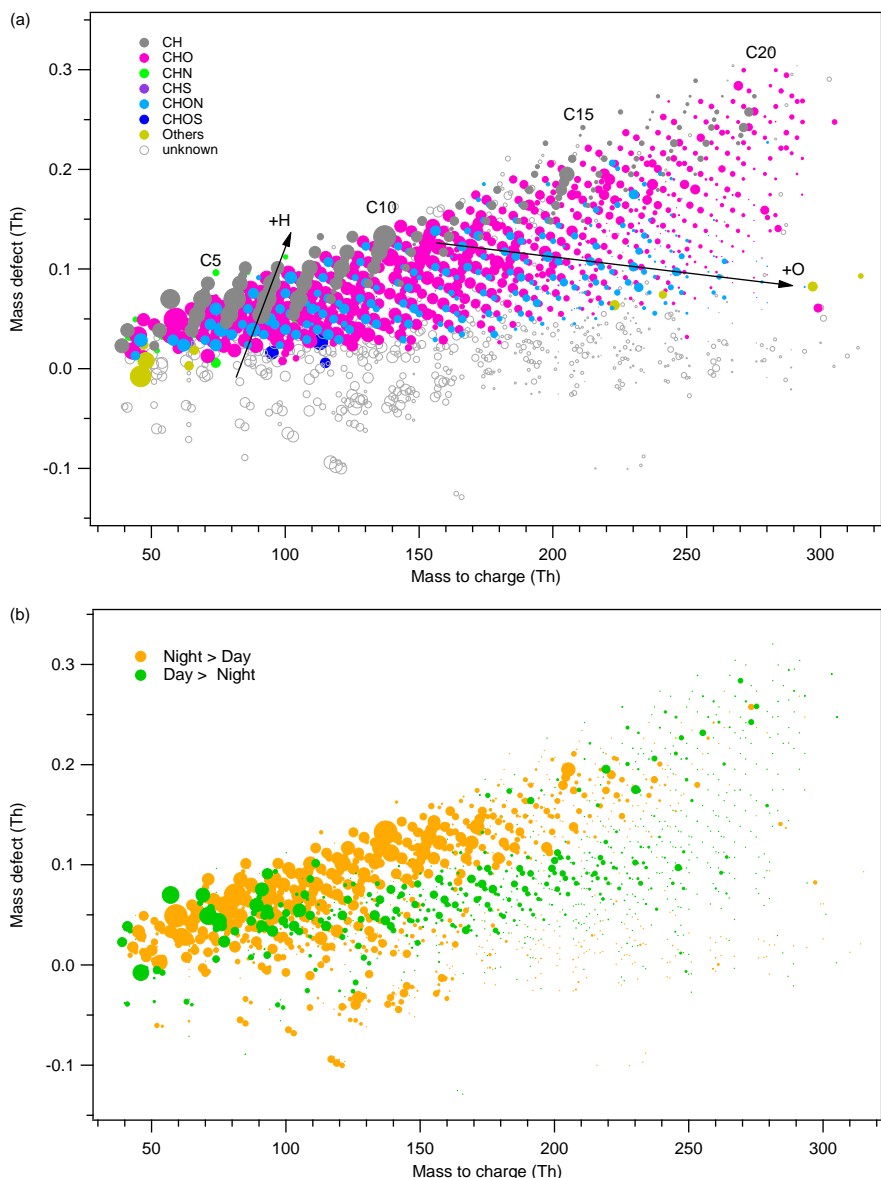

**Figure 2. Mass defect plot of the ions identified by high-resolution analysis of Vocus PTR-TOF data set. The x-axis shows the mass to charge ratio and the y-axis shows the mass defect, which is the deviation of the accurate mass from the nominal mass. Data points in (a) are color-coded by ion family (CH, CHO, CHN, CHS, CHON, CHOS) and sized by the logarithm of peak area. Data points in (b) are shown in orange when signals are higher during nighttime and in green when daytime signal is higher. The size corresponds to the difference of daytime and nighttime signal for the molecule.**



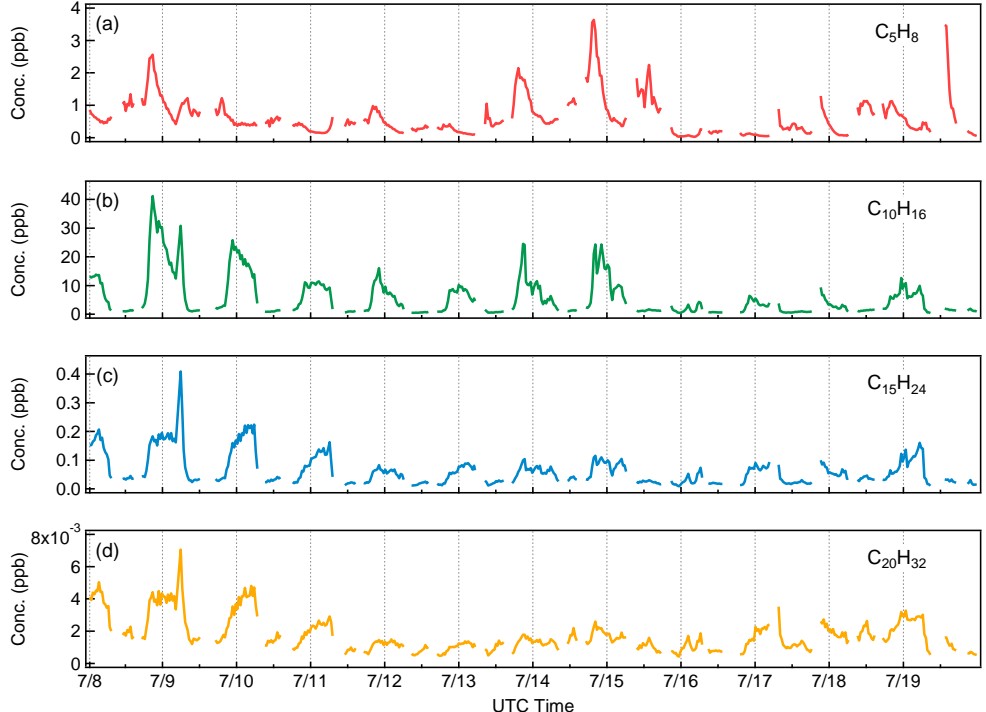

764

**Figure 3. Time series of (a) $C_5H_8$, (b) $C_{10}H_{16}$, (c) $C_{15}H_{24}$, and (d) $C_{20}H_{32}$.**





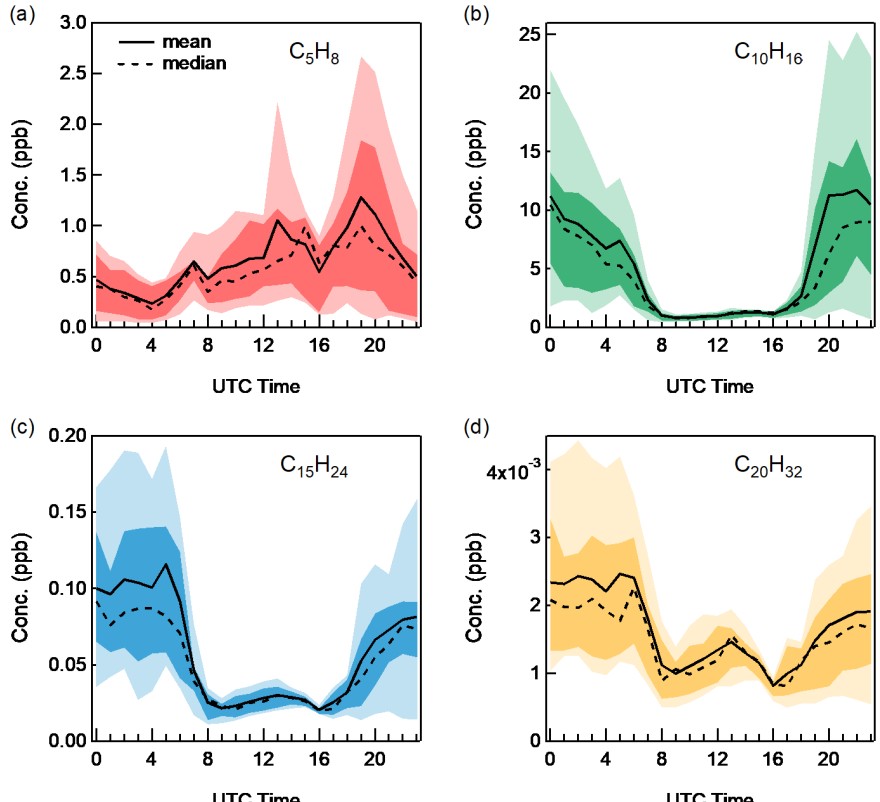

**Figure 4. Diurnal cycles of (a) C$_5$H$_8$, (b) C$_{10}$H$_{16}$, (c) C$_{15}$H$_{24}$, and (d) C$_{20}$H$_{32}$, with the 10th, 25th, 75th, and 90th percentiles shown in the shaded area.**





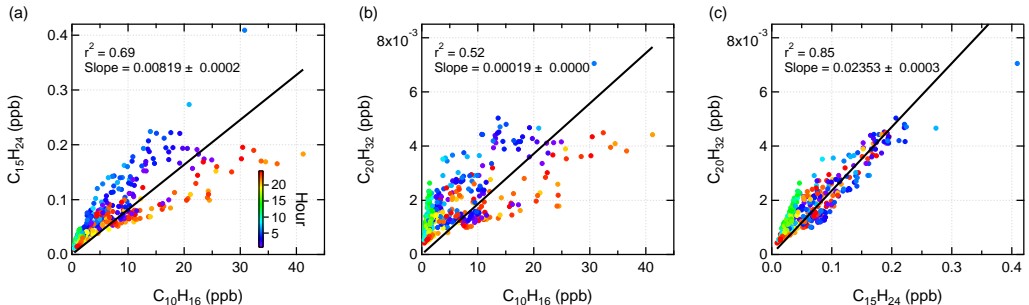


**Figure 5. Scatter plots of (a) $C_{15}H_{24}$ vs. $C_{10}H_{16}$, (b) $C_{20}H_{32}$ vs. $C_{10}H_{16}$, and (c) $C_{20}H_{32}$ vs. $C_{15}H_{24}$, colored by time of the**
**day.**





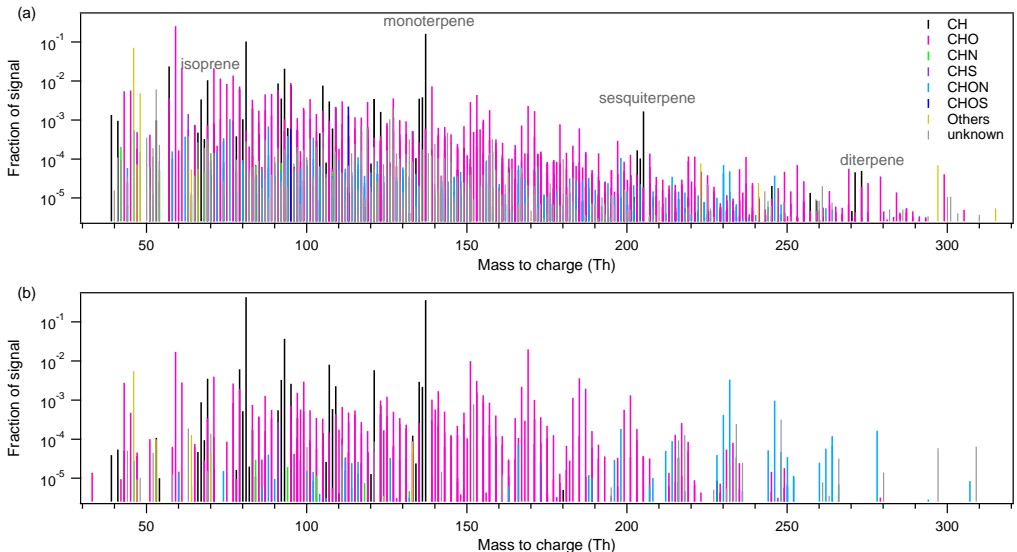


**Figure 6. Comparison of ambient average high-resolution mass spectra with those from α-pinene oxidation experiments**

**in the COALA chamber. (a) ambient observations in the Landes Forest; (b) α-pinene ozonolysis with NOₓ.**





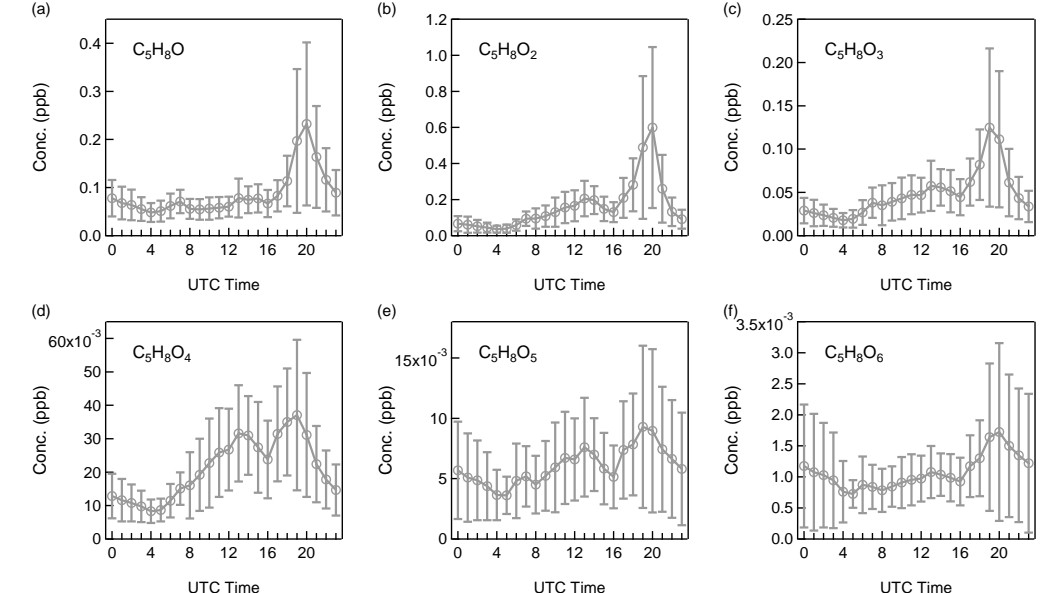

**Figure 7. Diurnal patterns of non-nitrate isoprene oxidation products: (a) $C_5H_8O$, (b) $C_5H_8O_2$, (c) $C_5H_8O_3$, (d) $C_5H_8O_4$, (e) $C_5H_8O_5$, and (f) $C_5H_8O_6$.**





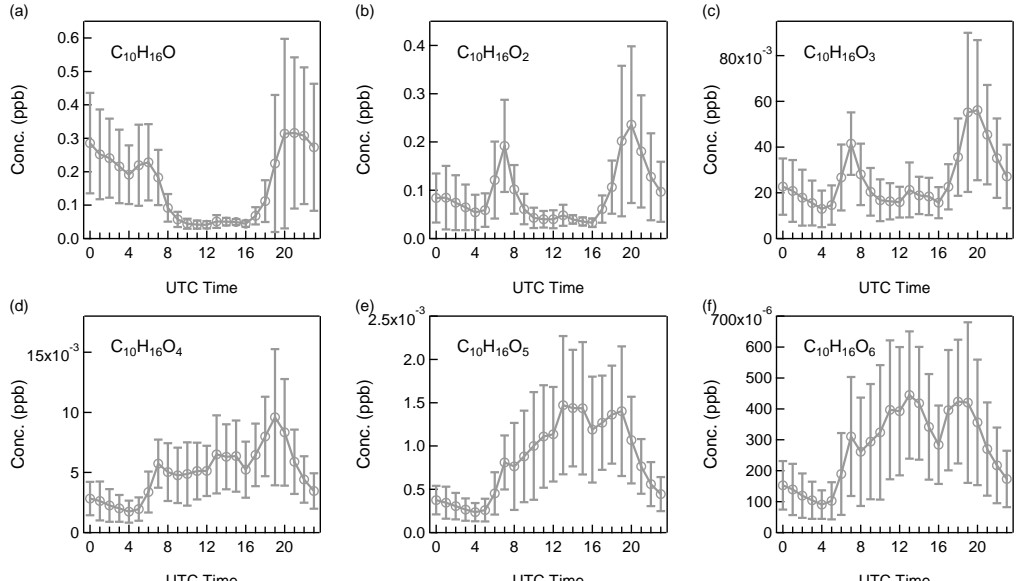


**Figure 8. Diurnal patterns of non-nitrate monoterpene oxidation products: (a) C$_{10}$H$_{16}$O, (b) C$_{10}$H$_{16}$O$_2$, (c) C$_{10}$H$_{16}$O$_3$, (d)**
**C$_{10}$H$_{16}$O$_4$, (e) C$_{10}$H$_{16}$O$_5$, and (f) C$_{10}$H$_{16}$O$_6$.**



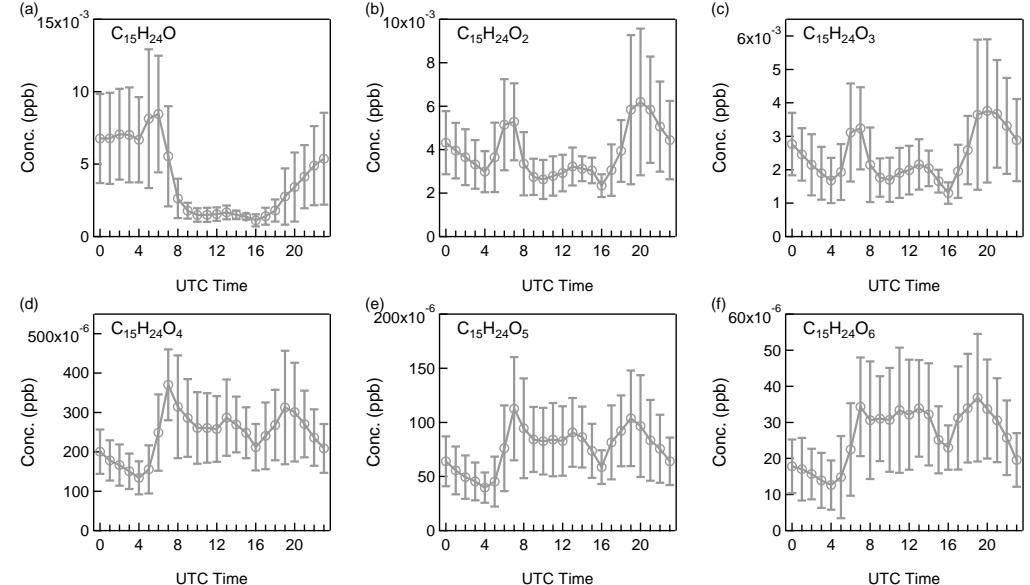


**Figure 9. Diurnal patterns of non-nitrate sesquiterpene oxidation products: (a) $C_{15}H_{24}O$, (b) $C_{15}H_{24}O_2$, (c) $C_{15}H_{24}O_3$, (d)**
**$C_{15}H_{24}O_4$, (e) $C_{15}H_{24}O_5$, and (f) $C_{15}H_{24}O_6$.**





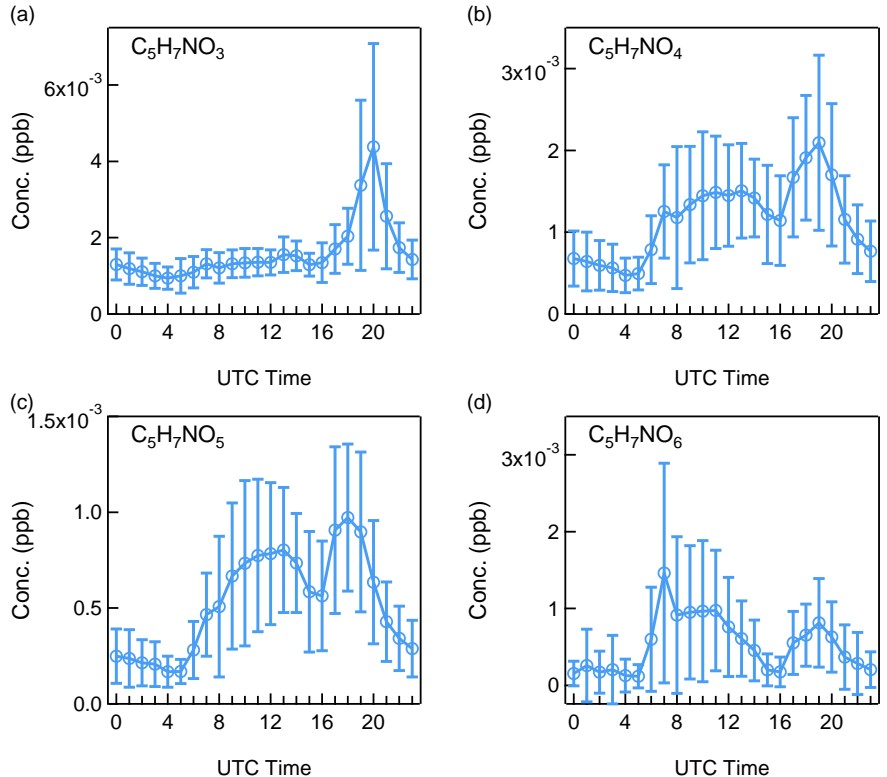


**Figure 10. Diurnal patterns of isoprene-derived organic nitrates: (a) C₅H₇NO₃, (b) C₅H₇NO₄, (c) C₅H₇NO₅, and (d)**


**C₅H₇NO₆.**





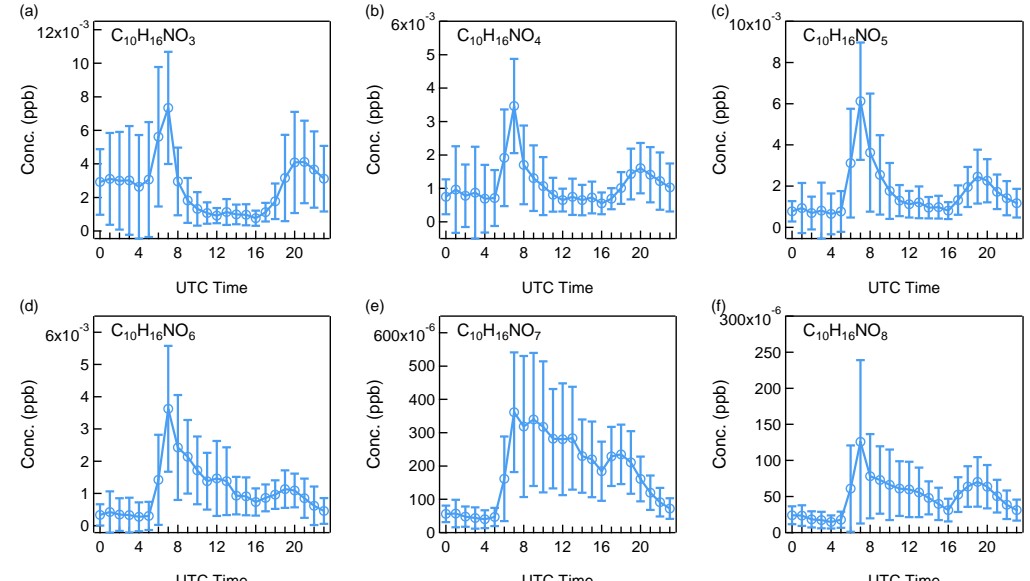


**Figure 11. Diurnal patterns of monoterpene-derived organic nitrates: (a) $C_{10}H_{15}NO_3$, (b) $C_{10}H_{15}NO_4$, (c) $C_{10}H_{15}NO_5$, (d)**
**$C_{10}H_{15}NO_6$, (e) $C_{10}H_{15}NO_7$, and (f) $C_{10}H_{15}NO_8$.**