# Peer review of "Terpenes and their oxidation products in the French Landes forest"

_Atmospheric Chemistry and Physics, 2019_

## Referee Comment (RC1) · Anonymous Referee #1 · 10 Oct 2019

The paper describes the results of organic trace gas measurements by Vocus PTR-TOF in the French Landes forest in the summer of 2018. The Vocus PTR-TOF is a newly developed PTR-MS instrument with improved detection limits and mass resolution, and the paper highlights the large number of compounds that can be detected with this instrument. In addition to the more commonly measured monoterpenes, the paper presents measurements of sesquiterpenes and diterpenes. The paper also illustrates that many oxidation products of these hydrocarbon precursors are detected. The analysis of the data is rather descriptive and focuses on diurnal variations and the potential importance of different oxidants in forming the observed products. Overall, the paper is suitable for publication in Atmospheric Chemistry and Physics after incorporation of

the comments below.

Section 2: A better description of the site is needed. Specifically, VOCs are sampled at 2 m height, which is well inside the canopy. How high are the treetops in this forest and how open is the canopy? Several studies have shown how strongly the mixing ratios of monoterpenes, light-dependent VOCs like MBO and their oxidation products can depend on height within the canopy (Holzinger et al., 2005). Some more discussion of the results in this context would be good to add to the paper.

Lines 38-40: The atmospheric chemistry of BVOCs has been studied much longer than just the past few years (Kanakidou et al., 2005). In general, the paper could do a better job citing the relevant literature. Much was learned about isoprene and monoterpene chemistry before the recent introduction of TOF-CIMS instruments.

Line 49: "irreversibly" instead of "irremediably".

Lines 54-56: The lack of sesquiterpene measurements by PTR-MS are mostly due to a lack of sensitivity.

Lack 60-62: I recommend adding a quantitative indication rather than "drastically enhanced" to describe how the sensitivities compare with other instruments.

Lines 97-99: I recommend adding the exact operating pressure in the reactor instead of a range. The difference between 1.0 and 1.5 mbar corresponds to a very large difference in E/N and therefore cluster ion distributions, fragmentation, etc.

Line 106: Have you considered how the use of nitrogen instead of zero air affects the ion chemistry and therefore background ion signals in the instrument?

Lines 117-118: This was not quite clear. Did your calibration mixture contain all three monoterpenes at 70 ppbv? In that case, your measured sensitivity is an average for the three monoterpenes. It is also not clear how you can use this average to determine the sensitivity as a function of reaction rate coefficient.

Lines 117-118: The lack of calibration for an oxygenated compound is a concern. The distribution of H3O+ and H3O+(H2O) reagent ions affects the sensitivity for hydrocarbons and polar molecules differently, but the distribution is difficult to determine in a Vocus PTR-TOF as H3O+ ions are very poorly detected. As the exact quantification of oxidation products is not a major focus in this study, I do not think it is a problem, but moving forward the Authors should consider calibrating their instrument for a much wider range in compounds.

Lines 120-122: I think this relationship needs to be included graphically. It seems that the range in rate coefficients is small and the resulting uncertainty in the factor 509.75 would consequently be very large.

Lines 123-125: It is trivial to determine the fragmentation of the monoterpenes in your calibration mixture and correct your measured sensitivities for fragmentation. This could be a large correction, so more detail needs to be given here.

Line 147: I do not think that inlet memory effects necessarily lead to an overestimate of sensitivities in this work. It all depends on how the passivation time of the inlet relates to the timescale of atmospheric variability. Memory effects can both lead to an underestimate and overestimate of measured mixing ratios.

Lines 163-164: It would take a lot of monoterpenes to consume 50 ppbv of ozone. Some back of the envelope estimate may be useful to constrain the chemical sink of ozone. Surface uptake is likely another important sink of ozone in the canopy.

Lines 177-179: Use "exact mass" instead of "accurate mass".

Lines 187-187: C6-C9 hydrocarbons are also notable. Some of these can be fragments of monoterpenes and sesquiterpenes. Also, the mass cut-off by the BSQ affects what can be seen below m/z ∼40 and the readers need to be made aware of that.

Lines 191-194: Biogenic butene is not very likely the cause for the elevated C4H9+ signal. As discussed, butanol is a more likely explanation. In addition, ions like C3H7+

and C4H9+ are very common fragments from many VOCs and are often prominent in the mass spectra (Pagonis et al., 2019).

Lines 215-216: Add the difference between UTC and local time. Given the diurnal variations in Fig. 4, the definition of day- and nighttime data seems a bit off.

Lines 224-235: I think the attribution of C5H9+ ions to isoprene should be considered in more detail. Isoprene mixing ratios are not very high in this study, and other VOCs are also detected at this mass. Notably, do the tree species at this site release MBO (Holzinger et al., 2005)?

Lines 268-274: Some further explanation of how the authors think monoterpenes could be detected as C15 and C20 is needed here.

Lines 290-292: The instrument settings used can indeed be the main explanation here and should be included in this paper.

Lines 315-317: Methyl vinyl ketone and methacrolein are the most common products from isoprene reactions with OH. The observation of C4 products does not necessarily imply ozone reactions.

Lines 327-337: This back-of-the-envelope analysis can be easily extended with estimates of the OH formation rate from alkene + ozone reactions, and the OH concentration in steady state. AN OH concentration of 10,000 seems very low.

Figure 2: Some indication of the low mass cut-off is needed to fully appreciate this graph: the Vocus PTR-TOF is less sensitive below m/z ~40 depending on the BSQ settings and many readers will not fully understand that. The colors used in panel b for day and night are hard to distinguish for the color blind. In the caption, use "exact mass" instead of "accurate mass".

References

Holzinger, R., Lee, A., Paw, K. T. and Goldstein, A. H.: Observations of oxidation

products above a forest imply biogenic emissions of very reactive compounds, Atmos. Chem. Phys., 5, 67–75, 2005.

Kanakidou, M., Seinfeld, J. H., Pandis, S. N., Barnes, I., Dentener, F. J., Facchini, M. C., Van Dingenen, R., Ervens, B., Nenes, A., Nielsen, C. J., Swietlicki, E., Putaud, J. P., Balkanski, Y. J., Fuzzi, S., Horth, J., Moortgat, G. K., Winterhalter, R., Myhre, C. E. L., Tsigaridis, K., Vignati, E., Stephanou, E. G. and Wilson, J.: Organic aerosol and global climate modelling: a review, Atmos. Chem. Phys., 5, 1053–1123., 2005.

Pagonis, D., Sekimoto, K. and de Gouw, J. A.: A library of proton-transfer reactions of H3O+ ions used for trace gas detection, J. Am. Soc. Mass Spectrom., 30, 1330–1335, doi:10.1007/s13361-019-02209-3, 2019.

---

## Referee Comment (RC2) · Anonymous Referee #2 · 16 Nov 2019

Li et al. show novel and interesting results of time-resolved chemical composition at a forested field site in France with a focus on terpenoids. This is one of early practical deployments of the novel VOCUS instrument. The paper is well written and nicely explores the impressive analytical capability of the instrument in its detection of terpenes and their oxidation products although in some places the story line has a high potential for improvement in story coherence and connection to process understanding and other PTRMS studies. I would have a few relatively minor comments but overall, I do not see an issue with recommending this overall nice paper after addressing my

comments.

**General**

C1) It is clear from an impressively large number of VOC ions that what is discussed is only a portion of a complex VOC mixture in this ecosystem. These types of super novel contributions are needed to make a step-change in the progress in understanding the full picture of atmospheric chemistry and physics. The low detection limit allows for detection of a dramatically larger number of ions including highly reactive and difficult to measure sesquiterpenes and diterpenes which are just example classes. Therefore I am surprised why the authors did not go for the broader embracement of the chemical composition because terpenes and terpenoids are not all the chemical families emitted by the forest. It should be possible to pick up all mVOCs, less common terpenoids including C-methylated terpenes such as homoterpenes (e.g. $C_{11}H_{18}$, $C_{16}H_{26}$), benzenoids and secondary metabolites, well known in chemical ecology.

C2) I am curious about chlorine radical chemistry of the forest terpenoids and the capability of detection of these products by VOCUS. Recent studies suggest that chlorine radical is more extensive than previously thought including noncoastal areas and for many VOCs it is much faster than other radicals (Wang and Hildebrandt-Ruiz, 2017).

**Specific**

C3) Abstract, L18, I was somewhat misled by elemental formula categories listed in the abstract. Are these really the only families detectable by VOCUS? What about halogenated, organometallic, and metaloorganic ions? Do you disregard the order of the elements in the formula? For example, HCNO and HNCO are completely different molecules. This way of elemental categories makes it unclear how many of each element in a molecule can be detected. It might be less distracting to just mention what elements can be in a detectable molecule or create a master formula (e.g. $C_{0-20}H_{0-42}O_{0-9}Si_{0-8}$...). What about inorganic compounds such as $H_2S$, $ClNH_2$?

C4) Abstract, L24, Why does the manuscript ignore an important Cl radical (e.g. Wang and Hildebrandt-Ruiz)?

C5) L30 what do you exactly mean by the relative term "ambient and remote"?

C6) L31 Why did the authors focus so much on oxidation in this field site? There must be beautiful primary emissions so the general question is how can we understand the oxidation process without understanding the underlying process of recognizing the full range of primary compounds? It is not just terpenes that get oxidized.

C7) L43 What about all the other primary hemiterpenoids, homoterpenes (in particular DMNT,TMTT), meroterpenes, and terpenoids that will get oxidized?

C8) L44 The formula of a diterpene is wrong here. Should be $C_{20}H_{32}$.

C9) L49 ULVOC is even less volatile than ELVOC (Schervish and Donahue, 2019).

C10) L55-56 There are more PTRMS papers which reported SQT (e.g. Bourtsoukidis et al., 2018).

C11) L99 The selection of the pressure range that is different from all the other CIMSes is unclear. Did you lower the pressure because the sensitivity was saturatingly too high or because you could not otherwise reach the desired E/N? What was the E/N ratio? If you ran only at a single E/N ratio, did you make an effort to optimize it for minimizing fragmentation of monoterpenes?

C12) Monoterpenes and sesquiterpenes fragment slightly differently at different E/N ratios (Misztal et al., 2013; Kim et al., 2012). The issue is that except for long-lived sesquiterpenes such as cedrene or copaene (note that these were not evaluated by Kim et al., 2012) majority of sesquiterpenes will fragment on the monoterpene parent and fragment ions. A similar issue might be with fragmentation of diterpenes on sesquiterpene ions. Have you thought about an algorithm to subtract the fragment contribution from higher terpenes? Given that VOCUS seems uniquely skilled in higher terpene detectability, it could be a simple calibration measurement with LCU using most

common isomers.

C13) L106 Did you use the completely dry N2 for background measurements? Although the sensitivities are not affected by ambient humidity, I am not sure it has been shown how stable the backgrounds are at different humidities. It is known that the methanol chemical background in PTRMS strongly depends on the humidity so the humidity of zero air should be carefully investigated.

C14) L122 I do not have an issue with the simplified empirical approach to derive sensitivities from k's as long as it is made clear that it is not generalizable to other conditions and instruments. In addition, I would expect the uncertainty is thoughtfully estimated and provided in the paper. However, this approach seems incorrectly applied to fragmenting compounds: "The predicted sensitivities with this method may be underestimated for compounds which do not fragment or fragment less than monoterpenes and cymene inside the PTR instruments." This does NOT make sense. One should sum up the known fragments and operate on the sum if the ions are pure and not interfering. It would be nice to see the monoterpene fragment distribution (e.g. Maleknia et al, 2007; Misztal et al., 2012) and if the sensitivity of the sum of fragments is consistent with the empirical k formula and explicit calibrations.

C15) L173. Could this result section title be rephrased to focus more on the science rather than the instrument?

C16) L190-203. I must admit that I was a little surprised why the terpenoid-oriented paper suddenly jumps into discussing so vigorously the unrejected C4 fragment and the speculation to its multi-identity suddenly weakens the otherwise strong story. Undoubtedly, it could be butene and/or butanol fragment (confirmed by spikes from the use of butanol at the site), and/or trans-hexenal emitted from wounded plants. What was not discussed is that it could also be a product of residual O2+ chemistry of alkanes (e.g. Amador-Munos et al., 2017). This points me to the more important point that it is unclear if the impurity ions were controlled or even checked for their relative

proportion to H3O+ ions? Apart from the C4H9+ ion, one would also expect C3H7+ and C5H11+ ions from the O2+ chemistry. In any case, it is distracting to focus on the C4H9+ ion so much in a terpenoid paper when you exclude from discussion hundreds of other probably more relevant and cleaner ions? I do not mean to criticize as it is overall a fair insight for the community but I would simply suggest moving this loose detail to SI to avoid unnecessary distraction.

C17) L208-2013 Again, why suddenly mention volatile siloxanes in a forest? I found it super distracting. Of course, VOCUS can detect these compounds as was already shown in Riva et al., 2019. The paper could make a connection to an observation that these compounds are present even in forested air far from human contributions but the sudden shift to this group of compounds can confuse readers about the sources. If you really want to make a connection, why not to refer to an idea that the signal could be used to evaluate anthropogenic contributions at the site or find leaks in the system? Otherwise it makes sense to delete this distracting fragment or move it to SI.

C18) I like the beautiful figures in this ms showing off the amazing capability of VOCUS. However, the science emanating from them is simply asking to be discussed more than superficially. The local time (UTC+1) would be better for a reader to avoid additional mental processing. Figure 4 axes and labels are inconsistently bolded. Figure 2 shows many potentially super interesting halogenated ions which are completely ignored in grey.

C19) The authors are in a great position to make a further insight into processes. For example, a better connection could be made with boundary layer dynamics responsible for diel trends of light-dependent isoprene vs other terpenes which can be emitted and accumulated at night (e.g. might consult Kaser et al., 2013 for a PTRTOF comparison). In terms of oxidation insights there are many papers which could be consulted in terms of the products and mechanisms (e.g. Lee et al., 2006, Kurten et al., 2017) and make an even better and more coherent connection to these valuable initial VOCUS field measurements.

**Technical**

C20) L61 "in" should be "of"

**References:**

Amador-Muñoz, O., Misztal, P. K., Weber, R., Worton, D. R., Zhang, H., Drozd, G., and Goldstein, A. H.: Sensitive detection of n-alkanes using a mixed ionization mode proton-transfer-reaction mass spectrometer, Atmos. Meas. Tech., 9, 5315–5329, https://doi.org/10.5194/amt-9-5315-2016, 2016.

Bourtsoukidis, E., Behrendt, T., Yañez-Serrano, A.M., Hellén, H., Diamantopoulos, E., Catão, E., Ashworth, K., Pozzer, A., Quesada, C.A., Martins, D.L. and Sá, M., 2018. Strong sesquiterpene emissions from Amazonian soils. Nature communications, 9(1), p.2226.

Lee, A., Goldstein, A.H., Keywood, M.D., Gao, S., Varutbangkul, V., Bahreini, R., Ng, N.L., Flagan, R.C. and Seinfeld, J.H., 2006. Gas‐phase products and secondary aerosol yields from the ozonolysis of ten different terpenes. Journal of Geophysical Research: Atmospheres, 111(D7).

Kaser, L., Karl, T., Guenther, A., Graus, M., Schnitzhofer, R., Turnipseed, A., Fischer, L., Harley, P., Madronich, M., Gochis, D. and Keutsch, E.N., 2013. Undisturbed and disturbed above canopy ponderosa pine emissions: PTR-TOF-MS measurements and MEGAN 2.1 model results.

Kurten, T., Møller, K.H., Nguyen, T.B., Schwantes, R.H., Misztal, P.K., Su, L., Wennberg, P.O., Fry, J.L. and Kjaergaard, H.G., 2017. Alkoxy radical bond scissions explain the anomalously low secondary organic aerosol and organonitrate yields from $\alpha$-pinene+ NO3. The journal of physical chemistry letters, 8(13), pp.2826-2834.

Maleknia, S.D., Bell, T.L. and Adams, M.A., 2007. PTR-MS analysis of reference and plant-emitted volatile organic compounds. International Journal of Mass Spectrometry, 262(3), pp.203-210.

Misztal, P.K., Heal, M.R., Nemitz, E. and Cape, J.N., 2012. Development of PTR-MS selectivity for structural isomers: Monoterpenes as a case study. International Journal of Mass Spectrometry, 310, pp.10-19.

Schervish, M. and Donahue, N. M.: Peroxy Radical Chemistry and the Volatility Basis Set, Atmos. Chem. Phys. Discuss., https://doi.org/10.5194/acp-2019-509, in review, 2019. Wang, D. S. and Ruiz, L. H.: Secondary organic aerosol from chlorine-initiated oxidation of isoprene, Atmos. Chem. Phys., 17, 13491–13508, https://doi.org/10.5194/acp-17-13491-2017, 2017.

---

## Author Comment (AC1) · 11 Jan 2020

**Response to Referee Comment 1 (RC1)**

The paper describes the results of organic trace gas measurements by Vocus PTRTOF in the French Landes forest in the summer of 2018. The Vocus PTR-TOF is a newly developed PTR-MS instrument with improved detection limits and mass resolution, and the paper highlights the large number of compounds that can be detected with this instrument. In addition to the more commonly measured monoterpenes, the paper presents measurements of sesquiterpenes and diterpenes. The paper also illustrates that many oxidation products of these hydrocarbon precursors are detected. The analysis of the data is rather descriptive and focuses on diurnal variations and the potential importance of different oxidants in forming the observed products. Overall, the paper is suitable for publication in Atmospheric Chemistry and Physics after incorporation of the comments below.

We thank the reviewer for the evaluation of the manuscript and the positive feedback. In the following, we answer the comments point by point and mention the changes that we made to our manuscript to address the reviewer's concerns and remarks.

Section 2: A better description of the site is needed. Specifically, VOCs are sampled at 2 m height, which is well inside the canopy. How high are the treetops in this forest and how open is the canopy? Several studies have shown how strongly the mixing ratios of monoterpenes, light-dependent VOCs like MBO and their oxidation products can depend on height within the canopy (Holzinger et al., 2005). Some more discussion of the results in this context would be good to add to the paper.

A more detailed description of the site has been added to the Section 2.1.

"Both population density and industrial emissions are low in this area. Due to the proximity of the Atlantic Ocean, the site has a strong maritime influence. The forest is largely composed of maritime pines (Pinus pinaster Aiton) and has an average height of ~10 m. Monoterpenes are known to be strongly emitted in the forest (Simon et al., 1994), which provides a good place for BVOCs characterization. A more detailed description of the site has been provided in earlier studies (Moreaux et al., 2011; Kammer et al., 2018; Bsaibes et al., 2019)."

Lines 38-40: The atmospheric chemistry of BVOCs has been studied much longer than just the past few years (Kanakidou et al., 2005). In general, the paper could do a better job citing the relevant literature. Much was learned about isoprene and monoterpene chemistry before the recent introduction of TOF-CIMS instruments.

We agree with the reviewer. More related literatures studying the atmospheric chemistry of BVOCs were added. "Over the past decades, a considerable amount of studies has been conducted to investigate the atmospheric chemistry of BVOCs (Kanakidou et al., 2005; Henze et al., 2006; Hatfield et al., 2011; Calfapietra et al., 2013; Jokinen et al., 2015; Ng et al., 2017)."

Line 49: "irreversibly" instead of "irremediably".

Changed.

Lines 54-56: The lack of sesquiterpene measurements by PTR-MS are mostly due to a lack of sensitivity.

It has been changed to "due to the relatively low sensitivity".

Lack 60-62: I recommend adding a quantitative indication rather than "drastically enhanced" to describe how the sensitivities compare with other instruments.

A quantitative description was added based on results from Holzinger et al. (2019). It has been changed to "with the enhanced sensitivities by a factor of ~10".

Lines 97-99: I recommend adding the exact operating pressure in the reactor instead of a range. The difference between 1.0 and 1.5 mbar corresponds to a very large difference in E/N and therefore cluster ion distributions, fragmentation, etc.

We agree with the reviewer. Generally, the Vocus ionization sources is operated at a low pressure. In this work, we operated it at a pressure of 1.5 mbar. The exact description has been added.

Line 106: Have you considered how the use of nitrogen instead of zero air affects the ion chemistry and therefore background ion signals in the instrument?

Zero air was not available at the field site so we used pure nitrogen which was also needed for other collocated measurements. As shown below (Fig. 1), the mass spectra remain quite similar between zero measurements using pure nitrogen and ambient measurements.

[Figure]

Figure 1. Example mass spectrum during (a) zero measurements using pure nitrogen and (b) ambient measurements.

Lines 117-118: This was not quite clear. Did your calibration mixture contain all three monoterpenes at 70 ppbv? In that case, your measured sensitivity is an average for the three monoterpenes. It is also not clear how you can use this average to determine the sensitivity as a function of reaction rate coefficient.

As described in the manuscript, the calibration mixture contains $m/z$ 137 (alpha/beta pinene + limonene) and $m/z$ 135 ($p$-cymene). For monoterpenes, yes, the measured sensitivity is an average for all these three monoterpenes. With the calculated sensitivities of monoterpenes and $p$-cymene and their rate constants, we built a linear regression of the sensitivity as a function of reaction rate coefficient.

Lines 117-118: The lack of calibration for an oxygenated compound is a concern. The distribution of H3O+ and H3O+(H2O) reagent ions affects the sensitivity for hydrocarbons and polar molecules differently, but the distribution is difficult to determine in a Vocus PTR-TOF as H3O+ ions are very poorly detected. As the exact quantification of oxidation products is not a major focus in this study, I do not think it is a problem, but moving forward the Authors should consider calibrating their instrument for a much wider range in compounds.

Thanks for the reviewer's suggestion. As the reviewer mentioned, the distribution of reagent ions affects the sensitivity of hydrocarbons and polar molecules differently. In the future, we consider calibrating the Vocus PTR-TOF with more oxygenated compounds, e.g., linanool oxide ($C_{10}H_{18}O_2$) and myrtenal ($C_{10}H_{14}O$).

Lines 120-122: I think this relationship needs to be included graphically. It seems that the range in rate coefficients is small and the resulting uncertainty in the factor 509.75 would consequently be very large.

As the reviewer suggested, the obtained empirical relationship has been included in the supplement as Figure S2. Due to the small difference between the rate coefficients of monoterpenes and p-cymene, we agree with the reviewer that the resulting uncertainty of the linear regression would be large. It has been noted in the revised manuscript: "Firstly, the small difference between the rate coefficients of monoterpenes and $p$-cymene may lead to large uncertainty in the established linear regression function between sensitivity and $k$. Calibrations with more VOC compounds should be performed in future works to cover a larger range of $k$ values."

[Figure]

Figure S2. The built empirical relationship between the sensitivities and the proton-transfer reaction rate coefficients ($k$) using the calibrated data of monoterpenes and $p$-cymene: Sensitivity (cps ppb$^{-1}$) = 828.9 $\times k$.

Lines 123-125: It is trivial to determine the fragmentation of the monoterpenes in your calibration mixture and correct your measured sensitivities for fragmentation. This could be a large correction, so more detail needs to be given here.

The study by Sekimoto et al. (2017) demonstrated that the sensitivity of VOCs is linearly correlated with the proton transfer reaction rate constant $k$, considering the ion transmission efficiency and the fragmentation of protonated VOCs inside the PTR instruments. Within the Vocus PTR-TOF, Krechmer et al. (2018) have shown that the transmission efficiencies of ions > $m/z$ 100 Th reach up to 99%. Therefore, to calculate sensitivities using the method of Sekimoto et al. (2017), the fragmentation correction should be included in this study.

Previous studies have shown that within the PTR instruments, protonated monoterpenes mainly produce fragment ions at $m/z$ 67, 81, and 95, and protonated $p$-cymene mainly produce fragment ions at $m/z$ 41, 91, 93, and 119 (Tani et al., 2003). According to our terpene calibrations, the residual fraction of protonated monoterpenes and $p$-cymene after fragmentation in the Vocus PTR-TOF was on average 66% and 55%, respectively. Therefore, the measured sensitivities of monoterpenes and $p$-cymene were corrected for fragmentation to build the linear regression between sensitivity and $k$. The updated plot showing the corrected sensitivities as a function of $k$ is displayed in Fig. S2. Detailed information has been added to the revised manuscript.

"Similar to conventional PTR instruments, the sensitivities of different VOCs in the Vocus PTR-TOF are linearly related to their proton-transfer reaction rate constants ($k$) when ion transmission efficiency and fragmentation ions are considered (Sekimoto et al., 2017; Krechmer et al., 2018). Krechmer et al. (2018) have shown that within the Vocus PTR-TOF, the transmission efficiencies of ions > $m/z$ 100 Th reach up to 99%. Therefore, the influence of fragmentation correction should be included in this study. According to terpene calibrations, the residual fraction was on average 66% and 55%, respectively, for protonated monoterpenes and $p$-cymene after their fragmentation within the instrument. Based on the corrected sensitivities for fragmentation and the $k$ values of monoterpenes and $p$-cymene, an empirical relationship between the sensitivity and $k$ was built from the scatterplots using linear regression: Sensitivity = 828.9 × $k$ (Fig. S2). Once $k$ is available, the sensitivity of a compound can be predicted. Some studies found that isoprene may fragment significantly to $m/z$ 41 (Keck et al., 2008; Schwarz et al., 2009). However, with the ambient data in this work, isoprene seems not to fragment much to $C_3H_5^+$, and they correlate poorly with each other (Fig. S3). Therefore, the fragmentation of isoprene is not considered for its quantification. Sesquiterpenes and some terpene oxidation products were found to fragment to varying degrees (Kim et al., 2009; Kari et al., 2018). Due to the lack of calibrations using other terpenes or terpene oxidation products, their fragmentation patterns within the Vocus PTR-TOF are not known in this work. Therefore, all the other terpenes and terpene oxidation products were quantified without consideration of fragment ions, which should be regarded as the lower limit of their ambient concentrations."

Line 147: I do not think that inlet memory effects necessarily lead to an overestimate of sensitivities in this work. It all depends on how the passivation time of the inlet relates to the timescale of atmospheric variability. Memory effects can both lead to an underestimate and overestimate of measured mixing ratios.

Losses of gas-phase compounds or delays on their transfer happen when they go through Teflon tubing or chambers (Pagonis et al., 2017; Deming et al., 2019). Delays on the transfer of these compounds cause memory effects and can lead to underestimation or overestimation of their concentrations. However, the losses of some gas-phase compounds onto tubing surface or chamber wall, especially those low-volatility compounds, can be irreversible. Therefore, due to their worse transmissions compared to the more volatile compounds, their sensitivities may be overestimated and thus their concentration can be underestimated.

Lines 163-164: It would take a lot of monoterpenes to consume 50 ppbv of ozone. Some back of the envelope estimate may be useful to constrain the chemical sink of ozone. Surface uptake is likely another important sink of ozone in the canopy.

Due to the higher monoterpene concentrations in the Landes forest (up to ~40 ppb at night), the chemical sink of ozone may be higher. But we agree with the reviewer that, in addition to the gas phase reactions of $O_3$ with terpenes, plants can also act as a sink for ozone through direct uptake. The explanation has been added to the revised manuscript.

"In addition, plant surface uptake is likely another important ozone sink in the canopy (Goldstein et al., 2004)."

Lines 177-179: Use "exact mass" instead of "accurate mass".

Changed.

Lines 187-187: C6-C9 hydrocarbons are also notable. Some of these can be fragments of monoterpenes and sesquiterpenes. Also, the mass cut-off by the BSQ affects what can be seen below m/z 40 and the readers need to be made aware of that.

The corresponding text has been modified in the revised manuscript.

"For hydrocarbons, multiple series with different carbon numbers were measured, especially those compounds containing 5 ("$C_5$") to 10 carbon atoms ("$C_{10}$"), 15 carbon atoms ("$C_{15}$"), and 20 carbon atoms ("$C_{20}$"). Some of the $C_5 – C_9$ ions can be fragments of monoterpenes, sesquiterpenes, and their oxidation products (Tani et al., 2003, 2013; Kim et al., 2009; Kari et al., 2018). For ions $< m/z$ 35 Th, the detection efficiency is considerably reduced due to the high-pass band filter of the BSQ (Krechmer et al., 2018)."

Lines 191-194: Biogenic butene is not very likely the cause for the elevated C4H9+ signal. As discussed, butanol is a more likely explanation. In addition, ions like C3H7+ and C4H9+ are very common fragments from many VOCs and are often prominent in the mass spectra (Pagonis et al., 2019).

Emissions of 1-butene have been measured in a midlatitude forest (Goldstein et al., 1996), a boreal wetland and forest floor (Hellén et al., 2006). Although in this study the biogenic butane does not likely explain the elevated $C_4H_9^+$ signal as well as the corresponding time variations, the readers should be aware of potential biogenic contributions in the forest.

We agree with the reviewer that $C_4H_9^+$ ions are very common fragment of many VOCs in PTR instruments and the corresponding explanation has been added to the revised manuscript.

"In addition, $C_4H_9^+$ ions are very common fragments of many VOCs in PTR instruments and the peaks are prominent in the mass spectra (Pagonis et al., 2019)."

Lines 215-216: Add the difference between UTC and local time. Given the diurnal variations in Fig. 4, the definition of day- and nighttime data seems a bit off.

The difference between UTC and local time is two hours, which has been added in the revised manuscript (Local time = UTC time + 2). The daytime and nighttime are defined based on the availability of sunlight. As shown in Fig. 1, we can check from the diurnal variations of solar radiation.

Lines 224-235: I think the attribution of C5H9+ ions to isoprene should be considered in more detail. Isoprene mixing ratios are not very high in this study, and other VOCs are also detected at this mass. Notably, do the tree species at this site release MBO (Holzinger et al., 2005)?

In the PTR instruments, the detected $C_5H_9^+$ ions can not only be isoprene but also fragments from many other compounds, i.e., cycloalkane and as mentioned by the reviewer 2-methyl-3-buten-2-ol (MBO). MBO undergoes collisional dissociation in the PTR and leads to the dominant fragment ion $C_5H_9^+$ (Karl et al., 2012). It has been shown that 71% of the parent ion of MBO fragments into $C_5H_9^+$ with an E/N ratio of 106 Td in a PTR-QMS (Warneke et al., 2003). At our measurement site, MBO was also detected by the Vocus PTR-TOF. However, due to the lack of MBO standards, we are not able to determine the fragmentation pattern of MBO within our instrument. As shown in the following figure, the $C_5H_9^+$ signal is around 10 times higher than $C_5H_{11}O^+$ signal. If $C_5H_{11}O^+$ ions largely contributed to $C_5H_9^+$, the correlation between these two ions is expected to be very good. However, as shown below, the correlation is weak (i.e., $r^2 = 0.33$). In addition, the diurnal variation of isoprene in Fig. 4a differs a lot with that of $C_5H_{11}O^+$ in Fig. S7. All this information demonstrates that the fragmentation of MBO does not have a significant influence on the attribution of $C_5H_9^+$ ions to isoprene in this study.

To make the readers aware of this, additional information has been added to the revised manuscript. "It has been shown that the attribution of $C_5H_9^+$ ions to isoprene with PTR instruments can be influenced by the fragmentation of many other compounds, i.e., cycloalkane and 2-methyl-3-buten-2-ol (MBO) (Karl et al., 2012; Gueneron et al., 2015). For example, using an *E/N* ratio of 106 Td in the PTR-MS with a quadrupole mass analyzer, 71% of the parent MBO fragmented to $C_5H_9^+$ ions (Warneke et al., 2003). However, in this study, the $C_5H_9^+$ signal was around 10 times as high as the $C_5H_{11}O^+$ signal and both ions correlated poorly with each other (Fig. S4; $r^2 = 0.33$). This information demonstrate that the fragmentation of MBO does not likely have a significant influence on the attribution of $C_5H_9^+$ ions to isoprene in this work."

[Figure]

Figure S4. Correlation of the time variations between $C_5H_{11}O^+$ and $C_5H_9^+$ signals.

Lines 268-274: Some further explanation of how the authors think monoterpenes could be detected as C15 and C20 is needed here.

A previous study shows that during pure isoprene oxidation experiments, ion signals at $m/z$ =137.133 ($C_{10}H_{17}^+$) and $m/z$ = 81.070 ($C_6H_9^+$) were detected by a PTR instrument (Bernhammer et al., 2018). These ion signals correspond to protonated monoterpenes and their major fragment. In this earlier study, two formation pathways of these signals were identified: secondary association reactions of protonated isoprene with isoprene within the PTR reaction chamber, and dimerization of pure isoprene inside the gas bottle to form monoterpenes. Similarly, in our ambient measurements, the detected $C_{15}$ and $C_{20}$ terpenes can possibly arise from the secondary association reactions of protonated monoterpenes with isoprene or monoterpenes respectively.

The corresponding explanation has been added in the revised manuscript.

"Bernhammer et al. (2018) have shown that secondary association reactions of protonated isoprene with isoprene can form monoterpenes within the PTR reaction chamber."

Lines 290-292: The instrument settings used can indeed be the main explanation here and should be included in this paper.

Different instrument settings, especially the varying $E/N$ ratios, can cause different fragmentation patterns of monoterpenes. As the $E/N$ value decreases, the percentage of fragment ions decreases because of the softer collisional reactions between $H_3O^+$ and monoterpene. However, in our ambient and chamber studies, the $E/N$ values of Vocus PTR-TOF were quite similar, 118 Td and 120 Td, respectively. To make it clear, we added the E/N values for ambient and chamber studies in the revised manuscript.

"In our ambient and chamber studies, the $E/N$ values of the Vocus PTR-TOF are quite similar, 118 Td and 120 Td, respectively."

Lines 315-317: Methyl vinyl ketone and methacrolein are the most common products from isoprene reactions with OH. The observation of C4 products does not necessarily imply ozone reactions.

We agree with the reviewer that isoprene ozonolysis, where one carbon is always split off from the molecule, is not the only way to form $C_4$ products. However, considering the peak concentration of isoprene and also the high ozone concentration at 8 pm in this study, isoprene ozonolysis is likely contributing to the formation of $C_4$ products in addition to isoprene reactions with OH. Additional information of isoprene reactions with OH to form MVK ($C_4H_6O$) and MACR ($C_4H_6O$) has been included in the revised manuscript.

"Reaction with OH represents the largest loss pathway for isoprene in the atmosphere and produces a population of isoprene peroxyl radicals (Wennberg et al., 2018). In the presence of NO, the major products are methyl vinyl ketone (MVK, $C_4H_6O$) and methacrolein (MACR, $C_4H_6O$)."

In addition, based on the competition between OH production and removal processes at night (Dusanter et al., 2008), the steady state OH concentration was estimated to be 0.012 ppt. With an $O_3$ concentration of ~20 ppb at 8 pm, the reaction rate of isoprene with OH radical was around 6 times as high as that of isoprene with $O_3$. Details can be found in the following response.

Lines 327-337: This back-of-the-envelope analysis can be easily extended with estimates of the OH formation rate from alkene + ozone reactions, and the OH concentration in steady state. AN OH concentration of 10,000 seems very low.

If the competition between OH production and removal processes lead to a steady state of OH formation, an estimation of OH concentration can be calculated using the following equation (Dusanter et al., 2008):

$$[OH]ss = \frac{k_{O3+VOC}\alpha[O3][alkene]}{k_{OH+VOC}[alkene] + k_{OH+O3}[O3]}$$

where $k_{O3+VOC}$ is the rate constant for O₃+alkene reaction with an OH yield of $\alpha$, $k_{OH+VOC}$ is the rate constant for OH+alkene reaction, $k_{OH+O3}$ is the rate constant for OH+ O₃ reaction. The rate constant of OH and O₃ reactions was obtained from Atkinson et al. (1992). At night, alkene concentrations in the Landes forest were dominated by monoterpenes, mainly α- and β-pinene (Riba et al., 1987; Simon et al., 1994). For the calculation of OH concentration, the loss of OH from reaction with O₃ was neglected, as it was much smaller than the loss of OH due to its reaction with monoterpenes (Gill and Hites, 2002). The rate constant of O₃ and monoterpene reactions was taken from Hakola et al. (2012), and the OH formation yield from O₃ and monoterpene reactions was obtained from Alicke et al. (2003). Finally, we assumed the equal contribution of α- and β-pinene to OH formation through alkene ozonolysis in this study. Hence, using an O₃ concentration of ~20 ppb at 8 pm, the OH concentration is estimated to be 0.012 ppt.

Details about the calculation of nighttime OH concentration from alkene ozonolysis have been added in the supplement. The corresponding text in the manuscript has been revised.

"Based on the competition between OH production and removal processes at night (Dusanter et al., 2008), the steady state OH concentration was estimated to be 0.012 ppt. Details can be found in the supplement. With an O₃ concentration of ~20 ppb at 8 pm, the reaction rate of isoprene with OH radical was around 6 times as high as that of isoprene with O₃."

Figure 2: Some indication of the low mass cut-off is needed to fully appreciate this graph: the Vocus PTR-TOF is less sensitive below m/z ~40 depending on the BSQ settings and many readers will not fully understand that. The colors used in panel b for day and night are hard to distinguish for the color blind. In the caption, use "exact mass" instead of "accurate mass".

It has been noted in the figure caption that ions $< m/z$ 35 Th are detected at a much-reduced efficiency due to a high-pass band filter in the BSQ. The colors in Figure 2b have been updated to be color blind friendly. "Accurate mass" was changed to "exact mass".

References

Holzinger, R., Lee, A., Paw, K. T. and Goldstein, A. H.: Observations of oxidation products above a forest imply biogenic emissions of very reactive compounds, Atmos. Chem. Phys., 5, 67–75, 2005.

Kanakidou, M., Seinfeld, J. H., Pandis, S. N., Barnes, I., Dentener, F. J., Facchini, M. C., Van Dingenen, R., Ervens, B., Nenes, A., Nielsen, C. J., Swietlicki, E., Putaud, J. P., Balkanski, Y. J., Fuzzi, S., Horth, J., Moortgat, G. K., Winterhalter, R., Myhre, C. E. L., Tsigaridis, K., Vignati, E., Stephanou, E. G. and Wilson, J.: Organic aerosol and global climate modelling: a review, Atmos. Chem. Phys., 5, 1053–1123., 2005.

Pagonis, D., Sekimoto, K. and de Gouw, J. A.: A library of proton-transfer reactions of H3O+ ions used for trace gas detection, J. Am. Soc. Mass Spectrom., 30, 1330–1335, doi:10.1007/s13361-019-02209-3, 2019.

Sekimoto, K., Li, S.-M., Yuan, B., Koss, A., Coggon, M., Warneke, C., and de Gouw, J.: Calculation of the sensitivity of proton-transfer-reaction mass spectrometry (PTR-MS) for organic trace gases using molecular properties, International Journal of Mass Spectrometry, 421, 71-94, https://doi.org/10.1016/j.ijms.2017.04.006, 2017.

Tani, A., Hayward, S., and Hewitt, C. N.: Measurement of monoterpenes and related compounds by proton transfer reaction-mass spectrometry (PTR-MS), International Journal of Mass Spectrometry, 223-224, 561-578, https://doi.org/10.1016/S1387-3806(02)00880-1, 2003.

Pagonis, D., Krechmer, J. E., de Gouw, J., Jimenez, J. L., and Ziemann, P. J.: Effects of gas–wall partitioning in Teflon tubing and instrumentation on time-resolved measurements of gas-phase organic compounds, Atmos. Meas. Tech., 10, 4687–4696, https://doi.org/10.5194/amt-10-4687-2017, 2017.

Deming, B. L., Pagonis, D., Liu, X., Day, D. A., Talukdar, R., Krechmer, J. E., de Gouw, J. A., Jimenez, J. L., and Ziemann, P. J.: Measurements of delays of gas-phase compounds in a wide variety of tubing materials due to gas–wall interactions, Atmos. Meas. Tech., 12, 3453–3461, https://doi.org/10.5194/amt-12-3453-2019, 2019.

Goldstein, A. H., Fan, S. M., Goulden, M. L., Munger, J. W., and Wofsy, S. C.: Emissions of ethene, propene, and 1-butene by a midlatitude forest, Journal of Geophysical Research: Atmospheres, 101, 9149-9157, 10.1029/96JD00334, 1996.

Hellén, H., Hakola, H., Pystynen, K.-H., Rinne, J., and Haapanala, S.: C2-C10 hydrocarbon emissions from a boreal wetland and forest floor, Biogeosciences, 3, 167–174, https://doi.org/10.5194/bg-3-167-2006, 2006.

Karl, T., Hansel, A., Cappellin, L., Kaser, L., Herdlinger-Blatt, I., and Jud, W.: Selective measurements of isoprene and 2-methyl-3-buten-2-ol based on NO+ ionization mass spectrometry, Atmos. Chem. Phys., 12, 11877–11884, https://doi.org/10.5194/acp-12-11877-2012, 2012.

Warneke, C., de Gouw, J. A., Kuster, W. C., Goldan, P. D., and Fall, R.: Validation of Atmospheric VOC Measurements by Proton-Transfer- Reaction Mass Spectrometry Using a Gas-Chromatographic Preseparation Method, Environmental Science & Technology, 37, 2494-2501, 10.1021/es026266i, 2003.

Bernhammer, A. K., Fischer, L., Mentler, B., Heinritzi, M., Simon, M., and Hansel, A.: Production of highly oxygenated organic molecules (HOMs) from trace contaminants during isoprene oxidation, Atmos. Meas. Tech., 11, 4763-4773, 10.5194/amt-11-4763-2018, 2018.

Dusanter, S., Vimal, D., and Stevens, P. S.: Technical note: Measuring tropospheric OH and HO2 by laser-induced fluorescence at low pressure. A comparison of calibration techniques, Atmos. Chem. Phys., 8, 321–340, https://doi.org/10.5194/acp-8-321-2008, 2008.

Atkinson, R., Baulch, D. L., Cox, R. A., Hampson, R. F., Kerr, J. A., and Troe, J.: Evaluated Kinetic and Photochemical Data for Atmospheric Chemistry: Supplement IV. IUPAC Subcommittee on Gas Kinetic Data Evaluation for Atmospheric Chemistry, Journal of Physical and Chemical Reference Data, 21, 1125-1568, 10.1063/1.555918, 1992.

Riba, M. L., Tathy, J. P., Tsiropoulos, N., Monsarrat, B., and Torres, L.: Diurnal variation in the concentration of α- and β-pinene in the landes forest (France), Atmospheric Environment (1967), 21, 191-193, https://doi.org/10.1016/0004-6981(87)90285-X, 1987.

Simon, V., Clement, B., Riba, M. L., and Torres, L.: The Landes experiment: monoterpenes emitted from maritime pine, J. Geophys. Res., 99, 16501–16510, 1994.

Gill, K. J., and Hites, R. A.: Rate Constants for the Gas-Phase Reactions of the Hydroxyl Radical with Isoprene, α- and β-Pinene, and Limonene as a Function of Temperature, The Journal of Physical Chemistry A, 106, 2538-2544, 10.1021/jp013532q, 2002.

Hakola, H., Hellén, H., Hemmilä, M., Rinne, J., and Kulmala, M.: In situ measurements of volatile organic compounds in a boreal forest, Atmos. Chem. Phys., 12, 11665-11678, 10.5194/acp-12-11665-2012, 2012.

Alicke, B., Geyer, A., Hofzumahaus, A., Holland, F., Konrad, S., Pätz, H. W., Schäfer, J., Stutz, J., Volz-Thomas, A., and Platt, U.: OH formation by HONO photolysis during the BERLIOZ experiment, Journal of Geophysical Research: Atmospheres, 108, PHO 3-1-PHO 3-17, 10.1029/2001JD000579, 2003.

**Response to Referee Comment 2 (RC2)**

Li et al. show novel and interesting results of time-resolved chemical composition at a forested field site in France with a focus on terpenoids. This is one of early practical deployments of the novel VOCUS instrument. The paper is well written and nicely explores the impressive analytical capability of the instrument in its detection of terpenes and their oxidation products although in some places the story line has a high potential for improvement in story coherence and connection to process understanding and other PTRMS studies. I would have a few relatively minor comments but overall, I do not see an issue with recommending this overall nice paper after addressing my comments.

We thank the reviewer for the evaluation of the manuscript and the positive feedback. In the following, we answer the comments point by point and mention the changes that we made to our manuscript to address the reviewer's concerns and remarks.

**General**

C1) It is clear from an impressively large number of VOC ions that what is discussed is only a portion of a complex VOC mixture in this ecosystem. These types of super novel contributions are needed to make a step-change in the progress in understanding the full picture of atmospheric chemistry and physics. The low detection limit allows for detection of a dramatically larger number of ions including highly reactive and difficult to measure sesquiterpenes and diterpenes which are just example classes. Therefore I am surprised why the authors did not go for the broader embracement of the chemical composition because terpenes and terpenoids are not all the chemical families emitted by the forest. It should be possible to pick up all mVOCs, less common terpenoids including C-methylated terpenes such as homoterpenes (e.g. $C_{11}H_{18}$, $C_{16}H_{26}$), benzenoids and secondary metabolites, well known in chemical ecology.

As shown in the manuscript, the Vocus PTR-TOF can detect large amounts of gas-phase signals in ambient deployment. It is difficult to characterize all the corresponding molecules within one publication. In addition, the main goal of this study is to demonstrate the capabilities of the recently developed Vocus PTR-TOF at measuring ambient air. To do this, terpenes were selected as the example because they are the main SOA precursor in the Landes forest, to propose a detailed analysis of their chemistry and highlight the performance of the Vocus PTR-TOF in characterizing atmospheric oxidation processes.

C2) I am curious about chlorine radical chemistry of the forest terpenoids and the capability of detection of these products by VOCUS. Recent studies suggest that chlorine radical is more extensive than previously thought including noncoastal areas and for many VOCs it is much faster than other radicals (Wang and Hildebrandt-Ruiz, 2017).

Theoretically, VOC compounds with higher proton affinity than $H_3O^+$ can be detected by Vocus PTR-TOF. However, the chloride-containing compounds have not been successfully identified in this study. It is unknown if chloride-containing compounds are detected during our measurements. Peaks with unidentified chemical formula are named as "unknown" in the mass defect plot. However, it is worth pointing out that the oxidation of alpha-pinene by chlorine atoms seems to proceed mainly through the H-atom abstraction as recently shown by Wang et al. (2019).

**Specific**

C3) Abstract, L18, I was somewhat misled by elemental formula categories listed in the abstract. Are these really the only families detectable by VOCUS? What about halogenated, organometallic, and

metaloorganic ions? Do you disregard the order of the elements in the formula? For example, HCNO and HNCO are completely different molecules. This way of elemental categories makes it unclear how many of each element in a molecule can be detected. It might be less distracting to just mention what elements can be in a detectable molecule or create a master formula (e.g. C0-20H0- 42O0-9Si0-8: : :). What about inorganic compounds such as H2S, ClNH2?

There are probably additional compounds with other elemental compositions that can be detected by Vocus. But they cannot be assigned with a specific elemental composition and are thus listed as "others" in this work. The mass spectrometric technique of PTR instruments allows separation of isobaric ions but not isomers. Therefore, the order of the elements is generally disregarded. As mentioned in the abstract, CH, CHO, CHN, CHS, CHON, CHOS, and others are listed to show what kinds of elemental composition categories are detected by the Vocus at the site. The specific number of elements in each category will vary a lot depending on the environmental conditions of the measurements. Halogenated, organometallic, and metaloorganic ions are not successfully identified in this study. The PTR instruments have been used to measure $H_2S$ in both laboratory and ambient environment. However, it was not detected in this work probably due to its very low concentration at this forest site. $H_2S$ has very bad ion transmission in the Vocus. As mentioned above, chloride-containing compounds are not successfully identified in this study. With the reviewer's suggestion, we checked the existence of $ClNH_2$ in the mass spectra and there seems to be a corresponding peak. But due to the interference of the high signals of $H_3O^+(H_2O)_2$ ($m/z$ 55 Th), the identification of $ClNH_2$ needs to be further evaluated.

C4) Abstract, L24, Why does the manuscript ignore an important Cl radical (e.g. Wang and Hildebrandt-Ruiz)?

The study by Wang and Hildebrandt-Ruiz (2017) investigated isoprene oxidation by Cl radicals. However, in this work, monoterpenes are the main SOA precursors in the Landes forest. As most of the oxidation products identified in this study do not contain any Cl, it is not possible to distinguish between Cl and OH-initiated oxidations. In addition, according to Wang et al. (2019), Cl-initiated oxidation of alpha-pinene does not produce much Cl-containing species.

C5) L30 what do you exactly mean by the relative term "ambient and remote"?

The ambient deployment of Vocus PTR-TOF was performed in a forested environment in this study, which is less influenced by anthropogenic sources. Hydrocarbon signals were dominated by monoterpenes. Therefore, the demonstrated capabilities of Vocus PTR-TOF were based on its performance in ambient and remote conditions in this work. The deployment of Vocus PTR-TOF in anthropogenic/polluted environment should be explored in future works.

C6) L31 Why did the authors focus so much on oxidation in this field site? There must be beautiful primary emissions so the general question is how can we understand the oxidation process without understanding the underlying process of recognizing the full range of primary compounds? It is not just terpenes that get oxidized.

A previous study by Kammer et al. (2018) suggests that terpene oxidations play an important role in SOA formation in the Landes forest. Therefore, the CERVOLAND campaign was organized to further assess the roles of BVOCs in aerosol formation at this forest site. During our measurements, monoterpene concentration reached up to 40 ppb at night and dominated the VOC emissions at this site. Therefore, terpene chemistry was investigated in this work as an example to achieve the goal.

C7) L43 What about all the other primary hemiterpenoids, homoterpenes (in particular DMNT,TMTT), meroterpenes, and terpenoids that will get oxidized?

The characteristics of some hemiterpenoids, i.e., prenol and isovaleric acid, have been illustrated in the manuscript and the supplement. For homoterpenes, DMNT was detected as a small peak during our measurements and TMTT was not clearly visible. The characteristics of some terpenoids, i.e., $C_{10}H_{16}O$ and $C_{10}H_{16}O_2$, were displayed in the manuscript and the supplement. However, it is true that not all BVOC compounds are investigated in this work. Since terpenes are characterized with much higher mixing ratios in the Landes forest, the oxidation processes of terpenes were demonstrated as the example to show the capabilities of the Vocus in atmospheric chemistry studies.

C8) L44 The formula of a diterpene is wrong here. Should be C20H32.

Corrected.

C9) L49 ULVOC is even less volatile than ELVOC (Schervish and Donahue, 2019).

Ultra-low volatility organic compounds (ULVOC) is a new class of organic products which is recently proposed by Schervish and Donahue et al. (2019). It has been added in the revised manuscript.

C10) L55-56 There are more PTRMS papers which reported SQT (e.g. Bourtsoukidis et al., 2018).

The ambient SQT measurements in Bourtsoukidis et al. (2018) were not performed with online PTR-MS but offline GC-MS.

C11) L99 The selection of the pressure range that is different from all the other CIMSes is unclear. Did you lower the pressure because the sensitivity was saturatingly too high or because you could not otherwise reach the desired E/N? What was the E/N ratio? If you ran only at a single E/N ratio, did you make an effort to optimize it for minimizing fragmentation of monoterpenes?

As described by Krechmer et al. (2018), the Vocus PTR-TOF is not a CIMS.

Before the ambient measurements, the instrument was carefully tuned for the optimal performance and minimize fragmentation of product ions. The E/N ratio was 118 Td during the campaign.

C12) Monoterpenes and sesquiterpenes fragment slightly differently at different E/N ratios (Misztal et al., 2013; Kim et al., 2012). The issue is that except for long-lived sesquiterpenes such as cedrene or copaene (note that these were not evaluated by Kim et al., 2012) majority of sesquiterpenes will fragment on the monoterpene parent and fragment ions. A similar issue might be with fragmentation of diterpenes on sesquiterpene ions. Have you thought about an algorithm to subtract the fragment contribution from higher terpenes? Given that VOCUS seems uniquely skilled in higher terpene detectability, it could be a simple calibration measurement with LCU using most common isomers.

It is true that sesquiterpenes will fragment on monoterpene parent and fragment ions to varying degrees based on the study by Kim et al. (2009). However, no standard calibration was available for sesquiterpenes and diterpenes in this work. Therefore, the quantification of sesquiterpenes and diterpenes may be underestimated and that of monoterpenes may be overestimated. It has been noted in the revised manuscript so that the readers are aware of that.

"Kim et al. (2009) show that different sesquiterpenes fragment on monoterpene parent and fragment ions to varying degrees inside the PTR instruments. Without the consideration of sesquiterpene fragmentation, the quantification of sesquiterpenes in this work may be underestimated."

In the future, the fragmentation of sesquiterpenes, diterpenes, and also some oxygenated compounds, inside the Vocus PTR-TOF should be investigated.

C13) L106 Did you use the completely dry N2 for background measurements? Although the sensitivities are not affected by ambient humidity, I am not sure it has been shown how stable the backgrounds are at different humidities. It is known that the methanol chemical background in PTRMS strongly depends on the humidity so the humidity of zero air should be carefully investigated.

Unlike other PTRMS, it has been shown that the sensitivity of the Vocus is independent of the relative humidity which is explained by the high concentration of water within the ion molecule reactor (Krechmer et al., 2018). Therefore, we do not expect to have a noticeable impact of the RH when measuring the background of the instruments with the Vocus. Though change in RH can impact the partitioning of gaseous species within the sampling line which is not tested during the blank measurements (i.e., injection of the clean air directly in the front of the FIMR).

C14) L122 I do not have an issue with the simplified empirical approach to derive sensitivities from k's as long as it is made clear that it is not generalizable to other conditions and instruments. In addition, I would expect the uncertainty is thoughtfully estimated and provided in the paper. However, this approach seems incorrectly applied to fragmenting compounds: "The predicted sensitivities with this method may be underestimated for compounds which do not fragment or fragment less than monoterpenes and cymene inside the PTR instruments." This does NOT make sense. One should sum up the known fragments and operate on the sum if the ions are pure and not interfering. It would be nice to see the monoterpene fragment distribution (e.g. Maleknia et al, 2007; Misztal et al., 2012) and if the sensitivity of the sum of fragments is consistent with the empirical k formula and explicit calibrations.

For both conventional PTR instrument and Vocus PTR-TOF, $k$ and sensitivity are linearly correlated. But the established relationship in this study is not applicable to other conditions or instruments. we have made it clear in the manuscript.

Detailed procedure was provided to derive the linear regression function between $k$ and sensitivities as well as potential uncertainty analysis. We agree with the reviewer that fragmentation of VOC compounds influence the derivation of the relationship. Therefore, as described in the response to referee #1, the fragmentation of monoterpenes and p-cymene inside the Vocus has been included. The influence of fragmentation on the quantification of other terpenes has also been discussed in the revised manuscript.

"Similar to conventional PTR instruments, the sensitivities of different VOCs in the Vocus PTR-TOF are linearly related to their proton-transfer reaction rate constants ($k$) when ion transmission efficiency and fragmentation ions are considered (Sekimoto et al., 2017; Krechmer et al., 2018). Krechmer et al. (2018) have shown that within the Vocus PTR-TOF, the transmission efficiencies of ions > $m/z$ 100 Th reach up to 99%. Therefore, the influence of fragmentation correction should be included in this study. According to terpene calibrations, the residual fraction was on average 66% and 55%, respectively, for protonated monoterpenes and $p$-cymene after their fragmentation within the instrument. Based on the corrected sensitivities for fragmentation and the $k$ values of monoterpenes and $p$-cymene, an empirical relationship between the sensitivity and $k$ was built from the scatterplots using linear regression: Sensitivity (cps ppb$^{-1}$) = 828.9 $\times k$ (Fig. S2). Once $k$ is available, the sensitivity of a compound can be predicted. It should be noted that the established relationship in this study is not applicable to other conditions or instruments. Some studies found that isoprene may fragment

significantly to *m/z* 41(Keck et al., 2008; Schwarz et al., 2009). However, with the ambient data in this work, isoprene seems not to fragment much to $C_3H_5^+$, and they correlate poorly with each other (Fig. S3). Therefore, the fragmentation of isoprene is not considered for its quantification. Sesquiterpenes and some terpene oxidation products were found to fragment to varying degrees (Kim et al., 2009; Kari et al., 2018). Due to the lack of calibrations using other terpenes or terpene oxidation products, their fragmentation patterns within the Vocus PTR-TOF are not known in this work. Therefore, all the other terpenes and terpene oxidation products were quantified without consideration of fragment ions, which should be regarded as the lower limit of their ambient concentrations."

C15) L173. Could this result section title be rephrased to focus more on the science rather than the instrument?

The major aim of this study is to demonstrate the Vocus PTR-TOF capabilities and highlight the importance of its applications in atmospheric sciences. It is important that Section 3.2 focuses more on the instrument to show the strong capability of the Vocus PTR-TOF as this study is the first report on its ambient measurements. Therefore, we would like to keep the title of Section 3.2 as it is.

C16) L190-203. I must admit that I was a little surprised why the terpenoid-oriented paper suddenly jumps into discussing so vigorously the unrejected C4 fragment and the speculation to its multi-identity suddenly weakens the otherwise strong story. Undoubtedly, it could be butene and/or butanol fragment (confirmed by spikes from the use of butanol at the site), and/or trans-hexenal emitted from wounded plants. What was not discussed is that it could also be a product of residual O2+ chemistry of alkanes (e.g. Amador-Munos et al., 2017). This points me to the more important point that it is unclear if the impurity ions were controlled or even checked for their relative proportion to H3O+ ions? Apart from the C4H9+ ion, one would also expect C3H7+ and C5H11+ ions from the O2+ chemistry. In any case, it is distracting to focus on the C4H9+ ion so much in a terpenoid paper when you exclude from discussion hundreds of other probably more relevant and cleaner ions? I do not mean to criticize as it is overall a fair insight for the community but I would simply suggest moving this loose detail to SI to avoid unnecessary distraction.

[Figure]

Figure 1. Example of the ambient mass spectra during the campaign, with a zoomed figure showing the relative proportion of $O_2^+$ and $H_3O^+$ ions.

For the impurity ions like $O_2^+$ and $NO^+$, we checked their relative proportion to $H_3O^+$ ions in the mass spectra. Due to the high-pass band filter in the BSQ, $O_2^+$, $NO^+$, and $H_3O^+$ are all detected at a much-reduced efficiency by the Vocus PTR-TOF. As shown above, the signal intensity of $H_3O^+$ ions is much higher than that of $O_2^+$ and $NO^+$ during our campaign. Therefore, the influence of impurity ions

can be neglected in this study. The residual $O_2^+$ chemistry will not have a big contribution to the detected $C_4H_9^+$ ions.

Finally, it is important to mention that $C_4H_9^+$ ranked the third largest peak in hydrocarbon signals. Therefore, we believed that it is important to discuss the detection of such ion by Vocus. As the reviewer suggested, the related discussion has been moved to the supplement.

C17) L208-2013 Again, why suddenly mention volatile siloxanes in a forest? I found it super distracting. Of course, VOCUS can detect these compounds as was already shown in Riva et al., 2019. The paper could make a connection to an observation that these compounds are present even in forested air far from human contributions but the sudden shift to this group of compounds can confuse readers about the sources. If you really want to make a connection, why not to refer to an idea that the signal could be used to evaluate anthropogenic contributions at the site or find leaks in the system? Otherwise it makes sense to delete this distracting fragment or move it to SI.

As the reviewer suggested, discussions related to volatile siloxanes have been deleted to avoid unnecessary distraction.

C18) I like the beautiful figures in this ms showing off the amazing capability of VOCUS. However, the science emanating from them is simply asking to be discussed more than superficially. The local time (UTC+1) would be better for a reader to avoid additional mental processing. Figure 4 axes and labels are inconsistently bolded. Figure 2 shows many potentially super interesting halogenated ions which are completely ignored in grey.

This study is the first one that reports the ambient deployment of the recently developed Vocus PTR-TOF. Therefore, the major aim of this study, as mentioned above, is to demonstrate the capabilities of the Vocus PTR-TOF and highlight its importance in atmospheric science studies. But we agree that more scientific information from the data set needs to be explored deeply in the future.

During the CERVOLAND campaign, data are recorded in UTC time for both Vocus PTR-TOF and all the other collocated instruments. Therefore, the data are presented in UTC time for a better and convenient comparison among all the measurements.

Figure 4 has been updated for the inconsistency.

Data points shown in grey in Figure 2 indicate those unidentified peaks. In this study, the halogenated ions are not successfully identified.

C19) The authors are in a great position to make a further insight into processes. For example, a better connection could be made with boundary layer dynamics responsible for diel trends of light-dependent isoprene vs other terpenes which can be emitted and accumulated at night (e.g. might consult Kaser et al., 2013 for a PTRTOF comparison). In terms of oxidation insights there are many papers which could be consulted in terms of the products and mechanisms (e.g. Lee et al., 2006, Kurten et al., 2017) and make an even better and more coherent connection to these valuable initial VOCUS field measurements.

Consulting to Kaser et al. (2013) and other references related to terpene emissions, a better connection was made between diel trends of terpenes and boundary layer dynamics. The corresponding information has been added in the revised manuscript.

"Isoprene emissions are strongly light-dependent (Monson et al., 1989; Kaser et al., 2013)."

"Different from the light-dependence of isoprene emissions, monoterpene emissions are found to be mainly controlled by temperature (Hakola et al., 2006; Kaser et al., 2013). At night, monoterpenes can be continuously emitted and accumulated within the boundary layer. Therefore, monoterpenes showed the opposite diel pattern to isoprene and peaked during nighttime."

We agree with the reviewer that the observations of terpenes and terpene oxidation products by Vocus PTR-TOF suggest complicated terpene chemical processes in the forest. However, as shown by Lee et al. (2006) and Kurten et al. (2017), laboratory simulations or theoretical computations are important to help figuring out the detailed chemical mechanisms. In addition to the Vocus ambient measurements, other data from collocated instruments, laboratory experiments, or theoretical simulations, are needed to provide a better figure of the complicated terpene chemical mechanisms, which is beyond the scope of this study. However, by evaluating the importance of different formation pathways in terpene chemistry in this study, we demonstrate the capability of the Vocus PTR-TOF at detecting of a wide range of oxidized reaction products and highlight the importance of its application in atmospheric science studies.

**Technical**

C20) L61 "in" should be "of"

Changed.

**References:**

Amador-Muñoz, O., Misztal, P. K., Weber, R., Worton, D. R., Zhang, H., Drozd, G., and Goldstein, A. H.: Sensitive detection of n-alkanes using a mixed ionization mode proton-transfer-reaction mass spectrometer, Atmos. Meas. Tech., 9, 5315–5329, https://doi.org/10.5194/amt-9-5315-2016, 2016.

Bourtsoukidis, E., Behrendt, T., Yañez-Serrano, A.M., Hellén, H., Diamantopoulos, E., Catão, E., Ashworth, K., Pozzer, A., Quesada, C.A., Martins, D.L. and Sá, M., 2018. Strong sesquiterpene emissions from Amazonian soils. Nature communications, 9(1), p.2226.

Lee, A., Goldstein, A.H., Keywood, M.D., Gao, S., Varutbangkul, V., Bahreini, R., Ng, N.L., Flagan, R.C. and Seinfeld, J.H., 2006. Gasâ˘AR˘ phase products and secondary aerosol yields from the ozonolysis of ten different terpenes. Journal of Geophysical Research: Atmospheres, 111(D7).

Kaser, L., Karl, T., Guenther, A., Graus, M., Schnitzhofer, R., Turnipseed, A., Fischer, L., Harley, P., Madronich, M., Gochis, D. and Keutsch, E.N., 2013. Undisturbed and disturbed above canopy ponderosa pine emissions: PTR-TOF-MS measurements and MEGAN 2.1 model results.

Kurten, T., Møller, K.H., Nguyen, T.B., Schwantes, R.H., Misztal, P.K., Su, L., Wennberg, P.O., Fry, J.L. and Kjaergaard, H.G., 2017. Alkoxy radical bond scissions explain the anomalously low secondary organic aerosol and organonitrate yields from α-pinene+ NO3. The journal of physical chemistry letters, 8(13), pp.2826-2834.

Maleknia, S.D., Bell, T.L. and Adams, M.A., 2007. PTR-MS analysis of reference and plant-emitted volatile organic compounds. International Journal of Mass Spectrometry, 262(3), pp.203-210.

Misztal, P.K., Heal, M.R., Nemitz, E. and Cape, J.N., 2012. Development of PTR-MS selectivity for structural isomers: Monoterpenes as a case study. International Journal of Mass Spectrometry, 310, pp.10-19.

Schervish, M. and Donahue, N. M.: Peroxy Radical Chemistry and the Volatility Basis Set, Atmos. Chem. Phys. Discuss., https://doi.org/10.5194/acp-2019-509, in review, 2019.

Wang, D. S. and Ruiz, L. H.: Secondary organic aerosol from chlorine-initiated oxidation of isoprene, Atmos. Chem. Phys., 17, 13491–13508, https://doi.org/10.5194/acp-17-13491-2017, 2017.

Wang, Y., Riva, M., Xie, H., Heikkinen, L., Schallhart, S., Zha, Q., Yan, C., He, X., Peräkylä, O., and Ehn, M.: Formation of highly oxygenated organic molecules from chlorine atom initiated oxidation of alpha-pinene, Atmos. Chem. Phys. Discuss., https://doi.org/10.5194/acp-2019-807, in review, 2019.

---

## Author Comment (AC2) · 11 Jan 2020

The comment was uploaded in the form of a supplement:
https://www.atmos-chem-phys-discuss.net/acp-2019-741/acp-2019-741-AC2-supplement.pdf